# Multiple Kronecker RLS fusion-based link propagation for drug-side effect prediction

**Yuqing Qian**                                                    *ustsyuqingqian@gmail.com*
*Institute of Fundamental and Frontier Sciences, University of Electronic Science and Technology of China, P.R.China*

**Ziyu Zheng**                                                    *smyzz16@nottingham.edu.cn*
*Department of Mathematical Sciences, University of Nottingham Ningbo, P.R.China*

**Prayag Tiwari***                                                 *prayag.tiwari@ieee.org*
*School of Information Technology, Halmstad University, Sweden*

**Yijie Ding***                                                    *wuxi_dyj@163.com*
*Yangtze Delta Region Institute (Quzhou), University of Electronic Science and Technology of China, P.R.China*

**Quan Zou**                                                       *zouquan@nclab.net*
*Institute of Fundamental and Frontier Sciences, University of Electronic Science and Technology of China, P.R.China*

*\*Corresponding author.*

**Reviewed on OpenReview:** *https://openreview.net/forum?id=LCPzaR9mML*

## Abstract

Drug-side effect prediction has become an essential area of research in the field of pharmacology. As the use of medications continues to rise, so does the importance of understanding and mitigating the potential risks associated with them. At present, researchers have turned to data-driven methods to predict drug-side effects. Drug-side effect prediction is a link prediction problem, and the related data can be described from various perspectives. To process these kinds of data, a multi-view method, called Multiple Kronecker RLS fusion-based link propagation (MKronRLSF-LP), is proposed. MKronRLSF-LP extends the Kron-RLS by finding the consensus partitions and multiple graph Laplacian constraints in the multi-view setting. Both of these multi-view settings contribute to a higher quality result. Extensive experiments have been conducted on drug-side effect datasets, and our empirical results provide evidence that our approach is effective and robust.

## 1 Introduction

Pharmacovigilance is critical to drug safety and surveillance. The field of pharmacovigilance plays a crucial role in public health by continuously monitoring and evaluating the safety profile of drugs. Pharmacovigilance involves collecting and analyzing data from various sources, including health care professionals (Yang et al., 2016), patients, regulatory authorities, and pharmaceutical companies. These data are then used to identify possible side effects and assess their severity and frequency (Da Silva & Krishnamurthy, 2016; Galeano et al., 2020). Traditionally, drug-side effects were primarily identified through spontaneous reporting systems, where health care professionals and patients reported adverse events to regulatory authorities. However, this approach has limitations, such as underreporting and delayed detection.

To overcome these limitations, researchers have turned to data-driven methods to find drug-side effects. With the advent of electronic health records, large-scale databases containing valuable information on medication usage and patient outcomes have become available. These databases have allowed researchers to analyze vast amounts of data to identify patterns between drugs and side effects.

One of the most commonly used approaches to drug-side effects prediction is model-based methods. Model-based methods involve the use of advanced statistical and machine learning techniques to extract knowledge from large datasets. By analyzing patterns in the data, researchers can identify potential drug-side effects and their associated risk factors. In their work, (Pauwels et al., 2011) predicted the side effects of drugs (Pau's method) by applying K-nearest neighbor (KNN), support vector machine (SVM), ordinary canonical correlation analysis (OCCA) and sparse canonical correlation analysis (SCCA) from drug chemical substructures; furthermore, their experiment outcome suggests that SCCA performs the best. Sayaka et al. (2012) utilized SCCA to associate targeted proteins with side effects (Miz's method). Liu et al. (2012) predicted drug side effects (Liu's method) using SVM and multivariate information, such as the phenotypic characteristics, chemical structures, and biological properties of the drug. Cheng et al. (2013) proposed a phenotypic network inference classifier to associate drugs with side effects (Cheng's method). NDDSA models (Shabani-Mashcool et al., 2020) the drug-side effects prediction problem using a bipartite graph and applies a resource allocation method to find new links. MKL-LGC (Ding et al., 2018) integrates multiple kernels to describe the diversified information of drugs and side-effects. These kernels are then combined using an optimized linear weighting algorithm. The Local and Global Consistency algorithm (LGC) is used to estimate new potential associations based on the integrated kernel information.

Deep learning techniques (Xu et al., 2022) have been increasingly used to predict drug side effects in recent years. These methods leverage the power of neural networks to analyze complex relationships between drugs, genes, and proteins. In SDPred (Zhao et al., 2022), chemical-chemical associations, chemical substructure, drug target information, word representations of drug molecular substructures, semantic similarity of side effects, and drug side effect associations are integrated. To learn drug-side effect pair representation vectors from different interaction maps, SDPred uses the CNN module. Drug interaction profile similarity (DIPA) provided the most contribution. GCRS (Xuan et al., 2022) builds a complex deep-learning structure to fuse and learn the specific topologies, common topologies and pairwise attributes from multiple drug-side effect heterogeneous graphs. Drug-side effect heterogeneous graphs are constructed using drug-side effect associations, drug-disease associations and drug chemical substructures. Based on a graph attention network, Zhao et al. (2021) developed a prediction model for drug-side effect frequencies that integrated information on similarity, known drug-side effect frequencies, and word embeddings. The above deep learning-based method is a kind of pairwise learning. To keep the sample balanced, this group selected the positive sample from trusted databases and the negative sample by random sampling. Such a treatment results in a certain loss of information and introduces noise to the label.

Drug-side effect prediction is a classic link prediction problem (Yuan et al., 2019). To solve this kind of problem, many multi-view methods have been proposed in recent years (Ding et al., 2021; 2016; Cichonska et al., 2018). Based on the information fusion at different stages of the training process, multi-view methods can roughly be divided into three categories: early fusion, late fusion and fusion during the training phase. Fig. 1 illustrates our taxonomy of multi-view learning method literature.

In early fusion techniques, the views are combined before training process is performed. Multiple kernel learning (MKL) (Wang et al., 2023b; Cichonska et al., 2018; Nascimento et al., 2016) is a typical early fusion technique. For each view, it computes one or more kernels, and then learns the optimal kernel from the base kernels. For example, MKL-KroneckerRLS (Ding et al., 2019) combines diversified information using Centered Kernel Alignment-based Multiple Kernel Learning (CKA-MKL). Based on the optimal kernel, Kronecker regularized least squares (Kro-RLS) was used to classify drug-side effect pairs. It must be noted that the performance of these methods relies heavily on the optimal view, which may be redundant or miss some key information. In late fusion techniques, a different model for each view is separately trained and later a weighted combination is taken as the final model. For instance, in Zhang et al. (2016), an ensemble model was constructed by integrating multiple methods, each providing a unique view. The model incorporates Liu et al. (2012), Cheng et al. (2013), a Integrated Neighbour-based Method (INBM), and a Restricted Boltzmann Machine-based Method (RBMBM). Each model is trained independently, and the final partition is the average weighted average of the base partitions. Late fusion allows for individual modeling of inherently different views, providing flexibility and advantage when dealing with diverse data. However, its drawback is the delayed coupling of information, limiting the extent to which each model can benefit from the information provided by other views.

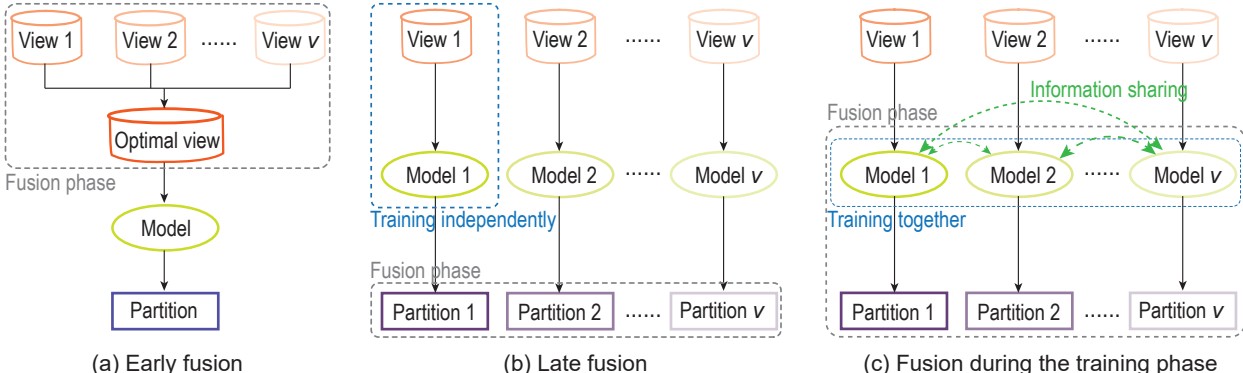

Figure 1: Taxonomy of multi-view learning framework literature. Note: "Partition" commonly refers to the learned result. This concept is more commonly found in classification and clustering tasks (Liu et al., 2023; Bruno & Marchand-Maillet, 2009; Wang et al., 2019). (a) Early fusion: the views are combined before the training process is performed; (b) Late fusion: a different model for each view is separately trained and then a combination is taken as the final partition; (c) Fusion during the training phase: it has some degree of freedom to model the views differently but to also ensure that information from other views is exploited during the training phase.

A third category is fusion during the training phase, which combines the benefits of both fusion types. It fuses multiple views at the partition level and enables the model to explore all views while being allowed to model one view differently. This framework has been applied to classification models (Houthuys & Suykens, 2021; Houthuys et al., 2018; Qian et al., 2022b; Xie & Sun, 2020) and clustering models (Lv et al., 2021; Houthuys et al., 2018; Wang et al., 2023a). By exploring consensus or complementarity information from multiple views, multi-view method can achieve better performance than single view method. The consensus principle pursues to achieve view-agreement among views. For instance, Wang et al. (2019) maximized the alignment between the consensus partition (clustering matrix) and the weighted combination of base partitions.

In this work, we apply this technique to the Kron-RLS algorithm. Due to its fast and scalable nature. The proposed method is named Multiple Kronecker RLS fusion-based link propagation (MKronRLSF-LP). Our work's main contributions are listed as follows:

(1) We extend Kron-RLS to the multiple information fusion setting by finding the consensus partition and multiple graph Laplacian constraint. Specifically, we generate multiple partitions by normal Kron-RLS and adaptively learn a weight for each partition to control its contribution to the shared partitions. This work was conducted with the aim of fusing partitions while still allowing for some flexibility in modeling single information. Furthermore, multiple graph Laplacian regularization is adopted to boost the performance of semi-supervised learning. Both settings co-evolve toward better performance.

(2) To fuse the features of multiple information more reasonably, we design an iterative optimization algorithm to effectively fuse multiple Kron-RLS submodels and obtain the final predictive model of drug-side effects. In the whole optimization, we avoid explicit computation of any pairwise matrices, which makes our method suitable for solving problems in large pairwise spaces.

(3) The proposed method can address the general link prediction problem; it is empirically tested on four real drug-side effect datasets, which are more sparse. The results show that MKronRLSF-LP can achieve excellent classification results and outperform other competitive methods.

The rest of this paper is organized as follows. Section 2 provides a description of the drug-side effect prediction problem. Section 3 reviews related work about MKronRLSF-LP. Section 4 comprehensively

presents the proposed MKronRLSF-LP. After reporting the experimental results in Section 5, we conclude this paper and mention future work in Section 6.

## 2 Problem description

Identification of drug-side effects is an example of the link prediction problem, which has the aim of predicting how likely it is that there is a link between two arbitrary nodes in a network. This problem can also be seen as a recommendation system (Jiang et al., 2019; Fan et al., 2021) task.

Let the drug nodes and side effect nodes of a network be $\mathbb{D} = \{d_1, d_2, \ldots, d_N\}$ and $\mathbb{S} = \{s_1, s_2, \ldots, s_M\}$, respectively. We denote the number of drug and side effect nodes by $N$ and $M$, respectively.

We define an adjacency matrix $\boldsymbol{F} \in \mathbb{R}^{N \times M}$ to represent the associations between drugs and side effects. Each element of $\boldsymbol{F}$ is defined as $\boldsymbol{F}_{i,j} = 1$ if the node pair $(d_i, s_j)$ is linked and $\boldsymbol{F}_{i,j} = 0$ otherwise.

The link prediction has the aim of predicting whether a link exists for the unknown state node pair $(d_i, s_j) \in \mathbb{D} \times \mathbb{S}$. Thus, it is a classification problem. Most methods use regression algorithms to predict a score (ranging from 0-1), which we call the link confidence. Then, a class of 0 or 1 is assigned to the predicted score by the threshold. Higher link confidence indicates a greater probability of the link existing, while lower values indicate the opposite. We define a new matrix $\hat{\boldsymbol{F}}$, which is estimated by the prediction model. Each of elements $\hat{\boldsymbol{F}}_{i,j}$ represents the predicted link confidence for the node pair $(d_i, s_j)$. Figure 5 summarizes the link prediction problem discussed in this paper.

## 3 Related work

### 3.1 Regularized Least Squares

The objective function of Regularized Least Squares (RLS) regression is:

$$\arg\min_{f} \ \frac{1}{2} \left\| \boldsymbol{F} - f\left(\boldsymbol{K}\right) \right\|_F^2 + \frac{\lambda}{2} \left\| f \right\|_K^2, \tag{1}$$

where $\lambda$ is a regularization parameter, $\|f\|_K$ denotes the RKHS norm (Kailath, 1971) of $f\left(\cdot\right)$. $f\left(\cdot\right)$ is the prediction function and be defined as:

$$f\left(\boldsymbol{K}\right) = \boldsymbol{K}\boldsymbol{a}, \tag{2}$$

where $\boldsymbol{a}$ is the solution of the model, $\boldsymbol{F}$ is a kernel matrix with elements

$$\boldsymbol{K}_{i,j} = k\left(d_i, d_j\right)\left(i, j = 1, \ldots, N\right), \tag{3}$$

and $k$ represents the kernel function.

By formulating the stationary points of Equation 1 and elimination the unknown parameters $\boldsymbol{a}$, the following solution is obtained

$$\hat{\boldsymbol{F}} = \boldsymbol{K}(\boldsymbol{K} + \lambda \boldsymbol{I}_N)^{-1} \boldsymbol{F}. \tag{4}$$

There is only one kind of feature space considered in this model. In the drug-side effect identification problem, there are two feature spaces: the drug space and the side effect space.

### 3.2 Kronecker Regularized Least Squares

Combining the kernels of the two spaces into a single large kernel that directly relates drug-side effect pairs would be a better option. Kronecker product kernel (Hue & Vert, 2010) is used for this. Given the drug kernel $\boldsymbol{K}_D$ and side effect kernel $\boldsymbol{K}_S$, then we have the kronecker product kernel

$$\boldsymbol{K} = \boldsymbol{K}_S \otimes \boldsymbol{K}_D, \tag{5}$$

where the $\otimes$ indicates the Kronecker product (Laub, 2004). By applying the Kronecker product kernel to RLS, the objective function of Kronecker Regularized Least Squares (Kron-RLS) is botained:

$$\arg\min_f \ \frac{1}{2}\left\|\text{vec}\left(\boldsymbol{F}\right) - f\left(\boldsymbol{K}\right)\right\|_F^2 + \frac{\lambda}{2}\left\|f\right\|_K^2, \tag{6}$$

where $\text{vec}\left(\cdot\right)$ is the vectorization operating function. By setting the derivative of Equation 6 $w.r.t$ $\boldsymbol{a}$ to zero, we obtain:

$$\boldsymbol{a} = \left(\boldsymbol{K} + \lambda\boldsymbol{I}_{NM}\right)^{-1}\text{vec}\left(\boldsymbol{F}\right). \tag{7}$$

Obviously, it needs calculating the inverse of $\left(\boldsymbol{K} + \lambda\boldsymbol{I}_{NM}\right)$ with size of $NM \times NM$, whose time complexity is $O\left(N^3M^3\right)$. Thus, a well-known theorem (Raymond & Kashima, 2010; Laub, 2004) is proposed to obtain the approximate inverse.

It is well known that the kernel (Liu et al., 2023; Pekalska & Haasdonk, 2008) matrices are positive semi-definite matrices, they can be eigen decomposed, $\boldsymbol{K}_D = \boldsymbol{V}_D\boldsymbol{\Lambda}_D\boldsymbol{V}_D^T$ and $\boldsymbol{K}_S = \boldsymbol{V}_S\boldsymbol{\Lambda}_S\boldsymbol{V}_S^T$. According to the theorem (Raymond & Kashima, 2010; Laub, 2004), the eigenvectors of the Kronecker product kernel $\boldsymbol{K}$ is the $\boldsymbol{V} = \boldsymbol{V}_S \otimes \boldsymbol{V}_D$. Define the matrix $\boldsymbol{\Lambda}$ to be either $\boldsymbol{\Lambda}_{i,j} = [\boldsymbol{\Lambda}_S]_{i,i} \times [\boldsymbol{\Lambda}_D]_{j,j}$. The eigenvalues of $\boldsymbol{K}$ is $\text{diag}\left(\text{vec}\left(\boldsymbol{\Lambda}\right)\right)$. The matrix $\boldsymbol{K} + \lambda\boldsymbol{I}_{NM}$ has the same eigenvactors $\boldsymbol{V}$, and eigenvalues $\text{diag}\left(\text{vec}\left(\boldsymbol{\Lambda} + \lambda\boldsymbol{1}\right)\right)$. Then, we can rewrite Equation 7 as:

$$\boldsymbol{K}(\boldsymbol{K} + \lambda\boldsymbol{I}_{NM})^{-1}\text{vec}\left(\boldsymbol{F}\right) = \boldsymbol{V}\text{diag}(\text{vec}\left(\boldsymbol{\Lambda}\right))\boldsymbol{V}^T\boldsymbol{V}\text{diag}(\text{vec}\left(\boldsymbol{\Lambda} + \lambda\boldsymbol{1}\right))^{-1}\boldsymbol{V}^T\text{vec}\left(\boldsymbol{F}\right). \tag{8}$$

Since $\boldsymbol{V}^T\boldsymbol{V} = \boldsymbol{I}_{NM}$ and $\text{diag}(\text{vec}\left(\boldsymbol{\Lambda}\right))\text{diag}(\text{vec}\left(\boldsymbol{\Lambda} + \lambda\boldsymbol{1}\right))^{-1}$ is also a diagonal matrix, we further simplify Equation 8 and get

$$\boldsymbol{K}(\boldsymbol{K} + \lambda\boldsymbol{I}_{NM})^{-1}\text{vec}\left(\boldsymbol{F}\right) = \boldsymbol{V}\text{diag}(\text{vec}\left(\boldsymbol{J}\right))\boldsymbol{V}^T\text{vec}\left(\boldsymbol{F}\right), \tag{9}$$

where the matrix $\boldsymbol{J}$ to be either

$$\boldsymbol{J}_{i,j} = \frac{\boldsymbol{\Lambda}_{i,j}}{\boldsymbol{\Lambda}_{i,j} + \lambda}. \tag{10}$$

Using the vec-tricks techniques $((\boldsymbol{A} \otimes \boldsymbol{B})\text{vec}\left(\boldsymbol{C}\right) = \text{vec}\left(\boldsymbol{B}\boldsymbol{C}\boldsymbol{A}^T\right))$, we further simplify Equation 8. Then, we get

$$\hat{\boldsymbol{F}} = \boldsymbol{V}_D\left(\boldsymbol{J} \odot \left(\boldsymbol{V}_D^T\boldsymbol{F}\boldsymbol{V}_S\right)\right)^T\boldsymbol{V}_S^T, \tag{11}$$

where $\odot$ represents the Hadamard product. The computational time of this optimization method is $O\left(N^3 + M^3\right)$, which is much less than $O\left(N^3M^3\right)$.

### 3.3 Kronecker Regularized Least Squares with Multiple Kernel Learning

Kron-RLS is a kind of kernel method. It can be difficult for nonexpert users to choose an appropriate kernel. To address such limitations, Multiple Kernel Learning (MKL) (Gönen & Alpaydın, 2011) is proposed. Since kernels in MKL can naturally correspond to different views, MKL has been applied with great success to cope with the multi-view data (Wang et al., 2021; Xu et al., 2021; Guo et al., 2021; Qian et al., 2022a; Wang et al., 2023b) by combining kernels appropriately.

Given predefined base kernels $\left\{\boldsymbol{K}_D^i\right\}_{i=1}^P$ and $\left\{\boldsymbol{K}_S^j\right\}_{j=1}^Q$ from drug feature space and side effect feature space, respectively. These kernels can be built from different types or views. The optimal kernel can be combined by a linear function corresponding to the base kernels:

$$\boldsymbol{K}_D^{opt} = \sum_{i=1}^P w^i\boldsymbol{K}_D^i. \tag{12}$$

Usually, an additional constraint is imposed on the corresponding combination coefficient $w$ to control its structure:

$$\sum_{i=1}^P w^i = 1, w^i \geq 0, i = 1, \ldots, P. \tag{13}$$

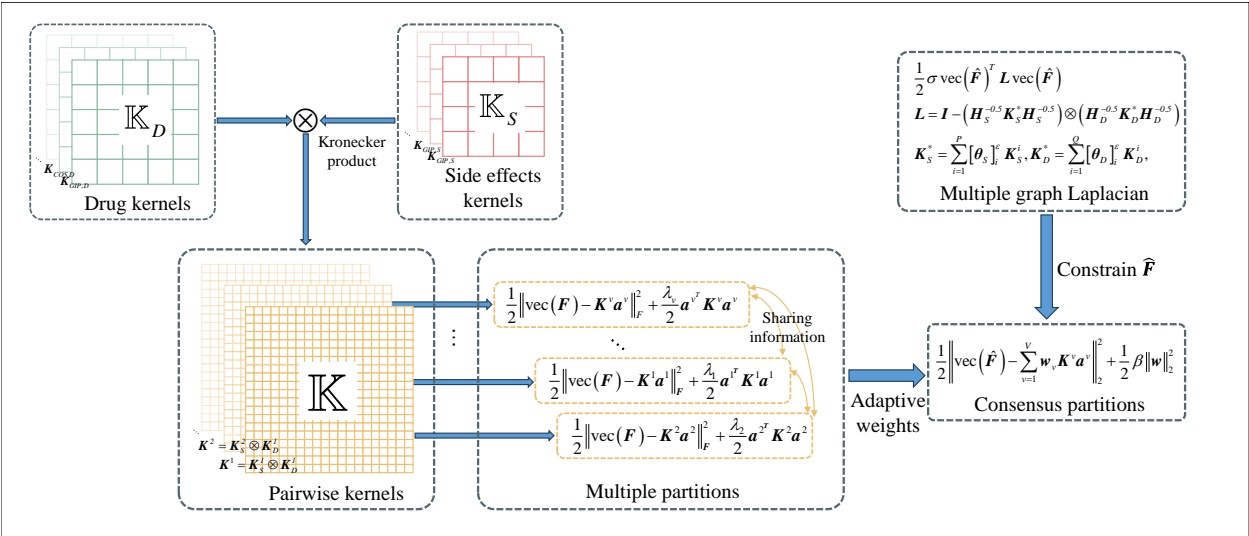

Figure 2: Framework diagram of MKronRLSF-LP. MKronRLSF-LP allow the multiple partitions have a degree of freedom to model the single information and introduce a multiple graph Laplacian regularization into consensus partition.

The optimal side effect kernel $\boldsymbol{K}_S^{opt}$ is omitted.

Based on MKL method, Ding et al. (2019) and Nascimento et al. (2016) developed Kron-RLS based MKL methods, called Kron-RLS with CKA-MKL and Kron-RLS with selfMKL, respectively. Kron-RLS with CKA-MKL combines diversified information using Centered Kernel Alignment-based Multiple Kernel Learning (CKA-MKL). In Kron-RLS with selfMKL, the weights indicating the importance of individual kernels are calculated automatically to select the more relevant kernels. The final decision function of both methods is given by:

$$\text{vec}\left(\hat{\boldsymbol{F}}\right) = \left(\boldsymbol{K}_S^{opt} \otimes \boldsymbol{K}_D^{opt}\right)\left(\boldsymbol{K}_S^{opt} \otimes \boldsymbol{K}_D^{opt} + \lambda \boldsymbol{I}_{NM}\right)^{-1}\text{vec}\left(\boldsymbol{F}\right). \tag{14}$$

## 4 Proposed method

Existing multi view fusion methods based on Kron-RLS all follow MKL framework. These methods optimize the optimal pairwise kernel as a linear combination of a set of base kernels. Prior to training, all views are fused, and information is not shared during training phase. This is typical early fusion technology. Our proposal addresses this limitation by fusing multi-view information in a consensus partition. Compared with MKL framework, the advantage of the proposed method is that it allows sub partitions to have a certain degree of freedom to model the single information. Further, multiple graph Laplacian regularization is introduced into the consensus partition to boost performance. Fig. 2 illustrates the main procedure of MKronRLSF-LP.

### 4.1 The construction of kernel matrix

Kron-RLS is a kind of kernel method. We construct drug kernels using five different kinds of functions.

Gaussian Interaction Profile (GIP):

$$\left[\boldsymbol{K}_{GIP,D}\right]_{i,j} = \exp\left(-\gamma\|d_i - d_j\|^2\right), \tag{15}$$

where $\gamma$ is the gaussian kernel bandwidth and $\gamma = 1$.

Cosine Similarity (COS):

$$[\boldsymbol{K}_{COS,D}]_{i,j} = \frac{d_i^T d_j}{|d_i| \, |d_j|}. \tag{16}$$

Correlation coefficient (Corr):

$$[\boldsymbol{K}_{Corr,D}]_{i,j} = \frac{\text{Cov}\,(d_i, d_j)}{\sqrt{\text{Var}\,(d_i)\,\text{Var}\,(d_j)}}. \tag{17}$$

Normalized Mutual Information (NMI):

$$[\boldsymbol{K}_{NMI,D}]_{i,j} = \frac{\text{Q}\,(d_i, d_j)}{\sqrt{\text{H}\,(d_i)\,\text{H}\,(d_j)}}, \tag{18}$$

where $\text{Q}\,(d_i, d_j)$ is the mutual information of $d_i$ and $d_j$. $\text{H}\,(d_i)$ and $\text{H}\,(d_j)$ are the entropies of $d_i$ and $d_j$, respectively.

Neural Tangent Kernel (NTK):

$$[K_{NTK,D}]_{i,j} = \mathbb{E}_{\theta \sim w}\left[ f_{NTK}\,(\theta, d_i), f_{NTK}\,(\theta, d_j) \right], \tag{19}$$

where $f_{NTK}$ is a fully connected neural network and $\theta$ is collection of parameters in this network.

Similarity, we construct the side effect kernels ($\boldsymbol{K}_{GIP,S}$, $\boldsymbol{K}_{COS,S}$, $\boldsymbol{K}_{Corr,S}$, $\boldsymbol{K}_{NMI,S}$, $\boldsymbol{K}_{NTK,S}$) in side effect space.

## 4.2 The MKronRLSF-LP model

Let us define two sets of base kernel sets separately:

$$\mathbb{K}_D = \left\{ \boldsymbol{K}_D^1, \ldots, \boldsymbol{K}_D^P \right\}, \tag{20a}$$

$$\mathbb{K}_S = \left\{ \boldsymbol{K}_S^1, \ldots, \boldsymbol{K}_S^Q \right\}, \tag{20b}$$

where $P$ and $Q$ represents the numbers of drug and side effect kernels, respectively. Based on the $\boldsymbol{K}_D$ and $\boldsymbol{K}_S$, we can get a set of pairwise kernels:

$$\mathbb{K} = \left\{ \boldsymbol{K}^1 = \boldsymbol{K}_S^1 \otimes \boldsymbol{K}_D^1, \ldots, \boldsymbol{K}^V = \boldsymbol{K}_S^P \otimes \boldsymbol{K}_D^Q \right\}, \tag{21}$$

where $V$ denotes the numbers of base pairwise kernels. Obviously, $V$ is equal to $P \times Q$.

By using multiple partitions, we can manipulate multiple views in a partition space, which enhances the robustness of the model. The following ensemble KronRLS model is obtained

$$\arg\min_{\boldsymbol{a}^v} \sum_{v=1}^{V} \left( \frac{1}{2} \left\| \text{vec}\,(\boldsymbol{F}) - \boldsymbol{K}^v \boldsymbol{a}^v \right\|_2^2 + \frac{\lambda_v}{2} \boldsymbol{a}^{v^T} \boldsymbol{K}^v \boldsymbol{a}^v \right). \tag{22}$$

In multi-view methods, the consensus principle establishes consistency between partitions from different views. However, it's essential to find that these partitions deliver varying degrees of importance to the final prediction, unlike fusion without discrimination. To facilitate this, we introduce a consensus partition, denoted by $\hat{\boldsymbol{F}}$. It is a weighted linear combination of partitions $\hat{\boldsymbol{F}}_v$ from multiple distinct views. A variable $\boldsymbol{w}_v$ is introduced for view $v$ which characterizes its importance, which is calculated based on the training error. To prevent sparse situations, we employ $\|\cdot\|_2^2$ to smooth the weights. Then, we have the following optimization problem

$$\arg\min_{\hat{\boldsymbol{F}}, \boldsymbol{a}^v, \boldsymbol{w}} \frac{1}{2} \left\| \text{vec}\,(\hat{\boldsymbol{F}}) - \sum_{v=1}^{V} \boldsymbol{w}_v \boldsymbol{K}^v \boldsymbol{a}^v \right\|_2^2 + \mu \sum_{v=1}^{V} \left( \frac{\boldsymbol{w}_v}{2} \left\| \text{vec}\,(\boldsymbol{F}) - \boldsymbol{K}^v \boldsymbol{a}^v \right\|_F^2 + \frac{\lambda_v}{2} \boldsymbol{a}^{v^T} \boldsymbol{K}^v \boldsymbol{a}^v \right) + \frac{1}{2}\beta \left\| \boldsymbol{w} \right\|_2^2$$

$$s.t. \sum_{v=1}^{V} \boldsymbol{w}_v = 1, \boldsymbol{w}_v \geq 0, v = 1, \ldots, V. \tag{23}$$

In Equation 23, we observe that the consensus partition $\hat{\boldsymbol{F}}$ fits to an adjacency matrix $\boldsymbol{F}$ by an indirect path. As described in section 2, false zeros represent unobserved links in the network. Hence, we must avoid overfitting the observed matrix $\boldsymbol{F}$. Inspired by manifold scenarios, the Laplacian operator adeptly mitigates overfitting and noise, preserving the original data structure and keeping nodes with common labels closely associated. This approach is simple, and empirical evidence confirms its effective performance (Pang & Cheung, 2017; Chao & Sun, 2019; Jiang et al., 2023). Here, we apply multiple graph Laplacian regularization to Equation 23, which can effectively explore multiple different views and boost the performance of $\hat{\boldsymbol{F}}$. Specifically, the Kronecker product Laplacian matrix is calculated from the optimal drug and side effect similarity matrix, which are weighted linear combinations of multiple related kernel matrices. The weight of each kernel can be adaptively optimized during the training process and reduce the impact of noisy or less relevant graphs. The optimization problems for MKronRLSF-LP can be formulated as:

$$
\begin{aligned}
\underset{\hat{\boldsymbol{F}},\boldsymbol{a}^v,\boldsymbol{w},\boldsymbol{\theta}_D,\boldsymbol{\theta}_S}{\arg\min} \quad & \frac{1}{2}\left\|\operatorname{vec}\left(\hat{\boldsymbol{F}}\right) - \sum_{v=1}^{V}\boldsymbol{w}_v\boldsymbol{K}^v\boldsymbol{a}^v\right\|_2^2 + \mu\sum_{v=1}^{V}\left(\frac{\boldsymbol{w}_v}{2}\left\|\operatorname{vec}\left(\boldsymbol{F}\right) - \boldsymbol{K}^v\boldsymbol{a}^v\right\|_2^2 + \frac{\lambda^v}{2}\boldsymbol{a}^{v^T}\boldsymbol{K}^v\boldsymbol{a}^v\right) + \frac{1}{2}\beta\left\|\boldsymbol{w}\right\|_2^2 \\
& + \frac{1}{2}\sigma\operatorname{vec}\left(\hat{\boldsymbol{F}}\right)^T\boldsymbol{L}\operatorname{vec}\left(\hat{\boldsymbol{F}}\right) \\
s.t. & \sum_{v=1}^{V}\boldsymbol{w}_v = 1, \boldsymbol{w}_v \geq 0, v=1,\ldots,V, \\
& \boldsymbol{L} = \boldsymbol{I}_{NM} - \left(\boldsymbol{H}_S^{-0.5}\boldsymbol{K}_S^*\boldsymbol{H}_S^{-0.5}\right)\otimes\left(\boldsymbol{H}_D^{-0.5}\boldsymbol{K}_D^*\boldsymbol{H}_D^{-0.5}\right), \\
& \boldsymbol{K}_S^* = \sum_{i=1}^{Q}[\boldsymbol{\theta}_S]_i^\varepsilon\boldsymbol{K}_S^i, \boldsymbol{K}_D^* = \sum_{i=1}^{P}[\boldsymbol{\theta}_D]_i^\varepsilon\boldsymbol{K}_D^i, \\
& \sum_{i=1}^{Q}[\boldsymbol{\theta}_S]_i = 1, [\boldsymbol{\theta}_S]_i \geq 0, i=1,\ldots,Q, \sum_{i=1}^{P}[\boldsymbol{\theta}_D]_i = 1, [\boldsymbol{\theta}_D]_i \geq 0, i=1,\ldots,P.
\end{aligned}
\tag{24}
$$

where $\boldsymbol{L}$ is a normalized laplacian matrix, $\boldsymbol{H}_S$ and $\boldsymbol{H}_D$ are diagonal matrix with the $j$th diagonal elements as $\sum_k[\boldsymbol{K}_S^*]_{j,k}$ and $\sum_k[\boldsymbol{K}_D^*]_{j,k}$, respectively. And, $\varepsilon > 1$, guaranteeing each graph has a particular contribution to the Laplacian matrix.

Due to the lack of space, we present optimization algorithm of the Equation 24 in Appendix Section A.1.

## 5 Experiments

In this section, the performance of MKronRLSF-LP is shown, and we make comparisons with baseline methods and other drug-side effect predictors.

### 5.1 Dataset

Table 1: Summary of the real drug-side effect datasets.

| Name | Drug | Side effect | Associations | Sparsity | Reference |
|------|------|-------------|--------------|----------|-----------|
| Liu | 832 | 1385 | 59205 | 94.86% | (Cheng et al., 2013) |
| Pau | 888 | 1385 | 61102 | 95.03% | (Pauwels et al., 2011) |
| Miz | 658 | 1339 | 49051 | 94.43% | (Sayaka et al., 2012) |
| Luo | 708 | 4192 | 80164 | 97.30% | (Luo et al., 2017) |

Four real drug-side effect datasets are used to assess the effectiveness of our proposed method. Pau dataset is derived from the SIDER database (Kuhn et al., 2010) which contains information about drugs and their recorded side effects. Miz dataset includes information about drug-protein interactions and drug-side effect interactions, obtained from the DrugBank (Wishart et al., 2006) and SIDER database, respectively. There were 658 drugs with both targeted protein and side effect information. Additionally, Liu et al. mapped drugs in SIDER to DrugBank 3.0 (Knox et al., 2010), resulting in a final dataset of 832 drugs and 1385

side effects. Luo dataset has a large number of side effects and was extracted from the SIDER 2.0. Table 1 summarizes information about the datasets. We can see that these four datasets are sparse. In other words, there are fewer positive samples than negative samples. Thus, drug-side effect prediction can be viewed as a classification problem with extremely imbalanced data.

## 5.2 Parament setting

In this paper, the objective function 24 contains the following regularization parameters: $\mu$, $\beta$, $\sigma$, $\varepsilon$ and $\lambda^v, v = 1, \ldots, V$. To find the right combinations of the regularization parameters of MKronRLSF-LP to give the best performance, the grid search method is performed on the Pau dataset. The optimal parameters with the best AUPR are selected.

We first select $\lambda^v, v = 1, \ldots, V$ by the relative pairwise kernel with a single view Kron-RLS model. For each parameter $\lambda^v$, we select it in the range from $2^{-5}$ to $2^5$ with step $2^1$. The optimal parameters $\lambda^v$ are shown in Table 4. According to a previous study(Shi et al., 2019), the performance is not affected by parameter $\varepsilon$, so it is set to 2. Then, we fix $\lambda^v, v = 1, \ldots, V$ at the best values and tune $\mu$, $\beta$, $\sigma$ from within the range $2^{-10}$ to $2^0$ with step $2^1$. The optimal regularization parameters are $\mu = 2^{-7}$, $\beta = 2^0$ and $\sigma = 2^{-8}$.

## 5.3 Baseline methods

In this work, we compare MKronRLSF-LP with the following baseline methods: BSV, Comm Kron-RLS(Perrone & Cooper, 1995), Kron-RLS+CKA-MKL(Ding et al., 2019), Kron-RLS+pairwiseMKL(Cichonska et al., 2018), Kron-RLS+self-MKL(Nascimento et al., 2016), MvGRLP(Ding et al., 2021) and MvGCN(Fu et al., 2022). Due to the lack of space, we present details of these baseline methods in Appendix Section A.3. For a fair comparison, the same input as our method is fed into these baseline methods. To achieve the best performance, we also adopt 5-fold CV on the Pau dataset to tune the parameters.

## 5.4 Threshold finding

Because the MKronRLSF-LP and baseline methods only output the value of regression, we apply a threshold finding operation. For a certain validation set in the five-fold cross-validation (5-fold CV) procedure, we collect the labels and their corresponding predicted scores. Then, we obtain the optimal threshold by maximizing the $F_{score}$ on the predicted scores and labels from this validation sets. A trend of $F_{score}$, *Recall* and *Precision* with different thresholds over four datasets is shown in Fig. 6. While the threshold of prediction rises, the values of *Recall* is rising. Oppositely, *Precision* is falling. The $F_{score}$ is the harmonic mean of the *Recall* and *Precision*. It thus symmetrically represents both *Recall* and *Precision* in one metric. Here, we find the optimal threshold under maximizing the value of $F_{score}$. Table 5 summarizes the thresholds of different baseline methods on different datasets.

## 5.5 Comparison with baseline methods

We conduct the 5-fold CV to evaluate the performance of our method versus the baseline method. To further provide a fair and comprehensive comparison, each algorithm is iterated 10 times with different cross index, and then the mean values and standard deviations are reported in Table 3. The best single view is $K_{GIP,D} \otimes K_{NTK,S}$, which is selected by 5-fold CV on Pau dataset.

First, we observe that the proposed method has the best AUPR and $F_{score}$ on all datasets. Especially, the proposed method has a higher AUPR and $F_{score}$ than BSV on datasets. This indicates the improvement in using multiple views. The simple coupling frameworks BSV and Comm perform well on the Pau dataset. However, BSV and Comm cannot perform as well on other datasets, which indicates that the simple fusion schemes are sensitive to the dataset and not robust. Furthermore, Kron-RLS+pairwiseMKL achieves the highest AUC of 95.01%, 95.02% and 94.70% on the Liu, Pau and Miz datasets, respectively. This shows slight improvements of 0.23%, 0.21% and 0.23% over our method, respectively. As we discussed in Section 5.1, drug-side effect prediction is an extremely imbalanced classification problem. The AUC can be considered

as the probability that the classifier will rank a randomly chosen positive instance higher than a randomly chosen negative instance. Therefore, the AUC is not an important metric for predicting drug side effects.

Another interesting observation is that MKronRLSF-LP outperforms other MKL strategy methods in comparison. For example, it exceeds the best MKL method (CKA-MKL) by 2.1%, 2.32%, 1.43%, 2.51% in terms of AUPR on Liu, Pau, Miz and Luo dataset, respectively. These results verify the effectiveness of the consensus partition and multiple graph Laplacian constraint.

For a more thorough analysis and reliable conclusions, we use post-hoc test statistics to statistically assess the different metrics shown in Table 3. Fig. 3 shows the results of these tests visualized as Critical Difference diagrams. These results show that MKronRLSF-LP is significantly better ranked than all methods in terms of AUPR, $Recall$ and $F_{score}$. In addition, MKronRLSF-LP is only inferior than Kron-RLS+pairwiseMKL and Kron-RLS+CKA-MKL in terms of AUC and $Precision$, respectively. Besides, MvGCN is worse ranked than our method. Another point worth mentioning is that there is no sufficient statistical evidence to support that MvGCN performs better than model-based methods. MvGCN uses shallow GCN to avoid over-smoothing. The shallow GCN (Miao et al., 2021) can only capture local neighbourhood information of nodes, but the global features of the network have not been fully explored. A result of this is inaccurate embedding vectors.

In summary, the above experimental results demonstrate the superior prediction performance of MKronRLSF-LP to other baseline methods. We attribute the superiority of MKronRLSF-LP as three aspects: (1) The consensus partition is derived through joint fusion of weighted multiple partitions; (2) MKronRLSF-LP utilizes the multiple graph Laplacian regularization to constrain the consensus predicted value $\hat{\boldsymbol{F}}$, which makes the consensus partition is robust; (3) Unlike existing MKL methods, the proposed MKronRLSF-LP fuses multiple pairwise kernels at the partition level. It is these three factors that contribute to the improvement in prediction performance.

## 5.6 Ablation study

To validate the benefits of jointly applying the consensus partition and multiple graph Laplacian constraint, we conduct an ablation study by excluding a particular component. First, we construct a Kron-RLS based on each pairwise kernel separately. Each partition learns independently, so it can be regarded as an ensemble Kron-RLS, and its objective function is Equation 22. The results should be consistent for each view, and heterogeneous views have varying degrees of importance in the final prediction. Therefore, we set a consensus partition $\hat{\boldsymbol{F}}$, which is a weighted linear combination of base partitions (as shown in Equation 23). To further improve the performance and robustness of the model, we apply multiple graph Laplacian constraints to the consensus partition. Finally, the objective function 24 of MKronRLSF-LP is obtained. The results of the ablation study are shown in Fig. 4. It can be observed that the consensus partition and the multiple graph Laplacian constraint is helpful for MKronRLSF-LP to achieve the best results.

## 5.7 Comparisons of computational speed

In order to demonstrate the effectiveness of MKronRLSF-LP, we are now comparing it to different baseline methods in terms of computational speed. Except MvGCN, other methods are performed on a PC equipped with an Intel Core i7-13700 and 16GB RAM. Because MvGCN is a deep learning-based method, it is performed on a workstation equipped with a NVIDIA GeForce RTX 3090 GPU. For all baseline methods, we tested 10 times to report the mean running time. The results are shown in Table 2. The results do not include the kernel calculation time.

As expected, learning from multiple views takes more time than learning from only one view (BSV). Also, since MKronRLSF-LP fuses multiple views at the partition level, it requires more running time than Kron-RLS+CKA-MKL and Kron-RLS+self-MKL. Another observation is that MKronRLSF-LP is much faster than Kron-RLS+pairwiseMKL. This can be explained by looking at the time complexity of MKronRLSF-LP and Kron-RLS+pairwiseMKL. The inverse of pairwise kernels dominates the time complexity of both methods. In our optimization algorithm, we use eigendecomposition techniques to compute the approximate inverse. The time complexity of our method is $O((P + I_{ter})N^3 + (Q + I_{ter})M^3)$. Dif-

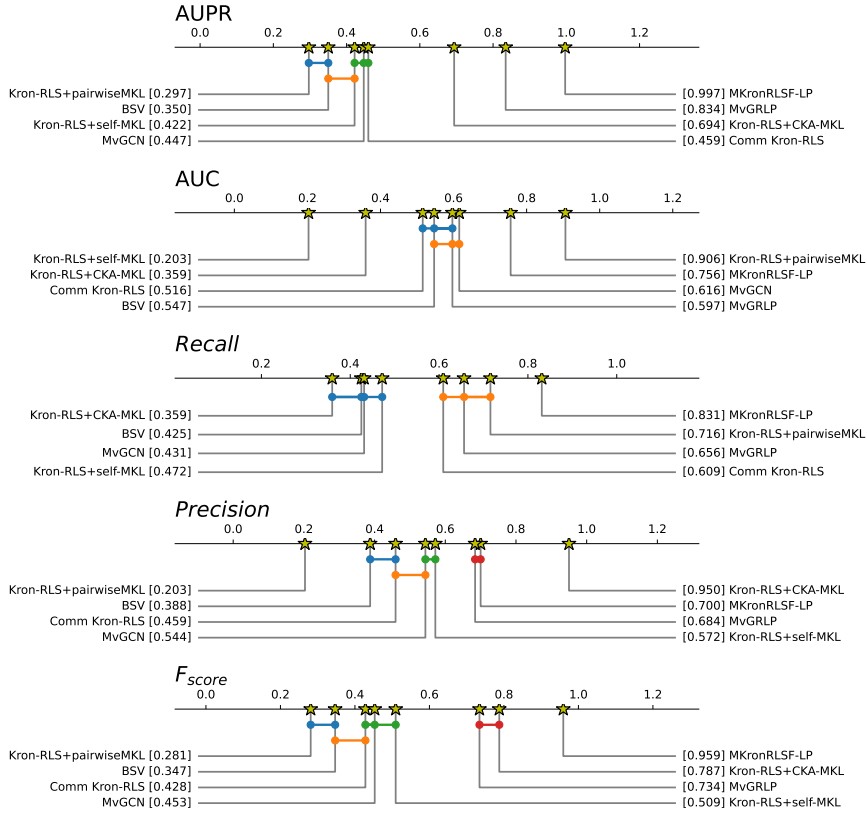

Figure 3: Critical difference diagram of average score ranks. A crossbar is over each group of methods that do not show a statistically significant difference among themselves.

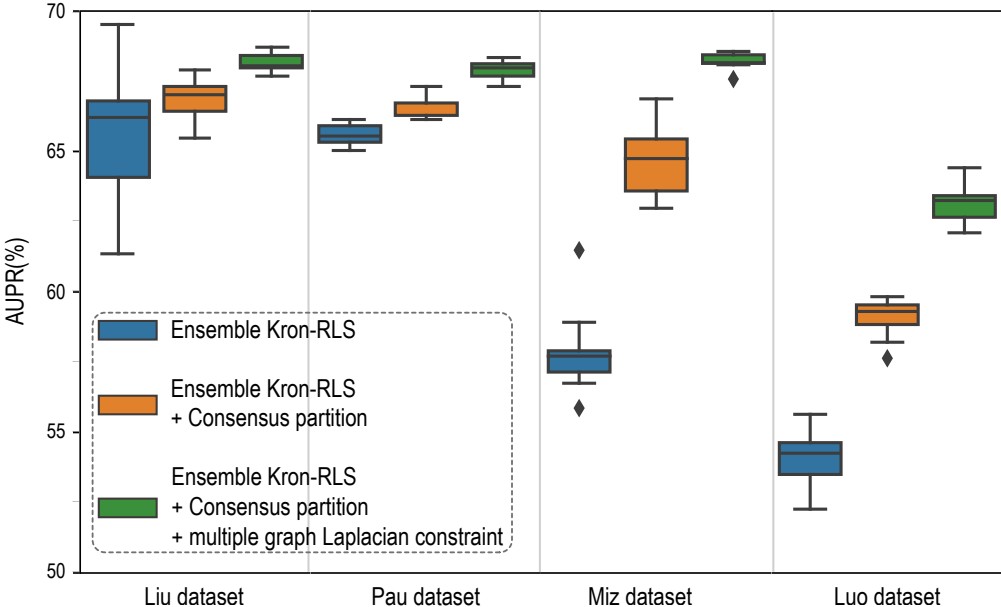

Figure 4: Ablation study of the consensus partition and multiple graph Laplacian constraint on four datasets.

Table 2: Mean running time (in seconds) of baseline methods on four datasets.

| Methods | Pau | Liu | Miz | Luo |
|---|---|---|---|---|
| BSV | 0.79 | 0.83 | 0.68 | 5.84 |
| Comm Kron-RLS | 19.38 | 20.95 | 18.39 | 148.60 |
| Kron-RLS+CKA-MKL | 2.69 | 2.18 | 2.36 | 13.13 |
| Kron-RLS+pairwiseMKL | 1583.67 | 1483.26 | 1364.21 | - |
| Kron-RLS+self-MKL | 12.21 | 13.05 | 12.09 | 155.85 |
| MvGRLP | 8.94 | 8.37 | 7.23 | 58.53 |
| MvGCN | 305.44 | 329.43 | 343.50 | - |
| MKronRLSF-LP | 50.55 | 43.9 | 35.83 | 280 |

- represents that the method took more than 2 hours to run.

ferently, Kron-RLS+pairwiseMKL solves the system with the conjugate gradient approach that iteratively improves the result by performing matrix-vector products. Hence, Kron-RLS+pairwiseMKL is carried out in $O(I_{ter}PQ(N^2M + M^2N))$. When MvGCN deal with Luo dataset, its running time exceeds 2 hours. This is because MvGCN utilizes a self-supervised learning strategy based on deep graph infomax (DGI) to initialize node embeddings. Whenever there are many nodes in a bipartite network, DGI takes a very long time to implement.

### 5.8 Comparison with other drug-side effect predictors

A comparison of the proposed drug-side effect prediction method with state-of-the-art methods is also provided. Tables 6,7, 8 and 9 present the results of 5-fold CV in terms of AUPR, AUC, $Recall$, $Precision$ and $F_{score}$ on the four datasets, respectively. We have highlighted the best results in bold and underlined the second-best results.

Obviously, MKronRLSF-LP achieves the highest AUPR and $F_{score}$ on all datasets. In the problem of drug-side effects prediction, AUPR and $F_{score}$ more desirable metrics (Ezzat et al., 2017; Li et al., 2021). Therefore, we conclude that our method outperforms the other assessed methods. GCRS (Xuan et al., 2022) and SDPred (Zhao et al., 2022) are deep learning-based methods. GCRS constructs multiple heterogeneous graphs and multi-layer convolutional neural networks with attribute-level attention to predict drug-side effect pair nodes. SDPred fuses multiple side information (including drug chemical structures, drug target, drug word, side effect semantic similarity, side effect word) by feature concatenation and adopts CNN and MLP for prediction tasks. However, on Luo dataset, GCRS and SDPred perform poorly; this is probably because they are pairwise learning methods and randomly negative sampling to construct the training set. The randomly negative sampling method cannot be guaranteed due to the reliability and quality of negative sample pairs, which results in a certain loss of information(Zhang et al., 2015; Ali & Aittokallio, 2019). The ensemble model (Zhang et al., 2016) combine Liu's method (Liu et al., 2012), Cheng's method (Cheng et al., 2013), INBM and RBM by the average scoring rule. It is obvious that the results of the ensemble model are significantly improved than the results of the sub-model on four datasets.

## 6 Conclusion

This paper presents MKronRLSF-LP for drug-side effect prediction. The MKronRLSF-LP method solves the general problem of multi-view fusion-based link prediction by utilizing the consensus partition and multiple graph Laplacian constraint. MKronRLSF-LP allows for some degree of freedom to model the views differently and combination weights for each view to find the consensus partition. Each view's weight is dynamically learned and plays a crucial role in exploring consensus information. It is found that the use of Laplacian regularization enhances semi-supervised learning performance, so a term of multiple graph Laplacian regularization is added to the objective function. Finally, we present an efficient alternating optimization algorithm. The results of our experiments indicate that our proposed methods are superior in terms of their classification results to other baseline algorithms and current drug-side effect predictors.

**Acknowledgments**

This work is supported in part by the National Natural Science Foundation of China (NSFC 62172076, 62250028 and U22A2038), the Zhejiang Provincial Natural Science Foundation of China (Grant No. LY23F020003), and the Municipal Government of Quzhou (Grant No. 2023D036).

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

# A Appendix

## A.1 Optimization

It is difficult and time-consuming to solve the Equation 24 because it contains multiple variables and large pairwise matrices. In this section, we divide the original problem into five subproblems and develop an iterative algorithm to optimize them. And, we avoid explicit computation of any pairwise matrices in the whole optimization, which makes our method suitable for solving problems in large pairwise spaces.

$\hat{\boldsymbol{F}}$-subproblem: we fix $\boldsymbol{a}^v$, $\boldsymbol{w}$, $\boldsymbol{\theta}_D$ and $\boldsymbol{\theta}_S$ to optimize variants $\hat{\boldsymbol{F}}$. Let $\boldsymbol{A} = \boldsymbol{H}_S^{-0.5} \boldsymbol{K}_S^* \boldsymbol{H}_S^{-0.5}$, $\boldsymbol{B} = \boldsymbol{H}_D^{-0.5} \boldsymbol{K}_D^* \boldsymbol{H}_D^{-0.5}$ and $\text{vec}\left(\hat{\boldsymbol{F}}^v\right) = \boldsymbol{K}^v \boldsymbol{a}^v$. Then, the optimization model of $\hat{\boldsymbol{F}}$ as follows:

$$\arg\min_{\hat{\boldsymbol{F}}} \frac{1}{2} \left\| \text{vec}\left(\hat{\boldsymbol{F}}\right) - \sum_{v=1}^V \boldsymbol{w}_v \text{vec}\left(\hat{\boldsymbol{F}}^v\right) \right\|_2^2 + \frac{1}{2}\sigma \text{vec}\left(\hat{\boldsymbol{F}}\right)^T \boldsymbol{L} \text{vec}\left(\hat{\boldsymbol{F}}\right) \tag{25}$$
$$s.t. \boldsymbol{L} = \boldsymbol{I}_{NM} - \boldsymbol{A} \otimes \boldsymbol{B}.$$

Let the derivative of Equation 25 *w.r.t* $\hat{\boldsymbol{F}}$ to zero, the solution of $\hat{\boldsymbol{F}}$ can be obtained:

$$\text{vec}\left(\hat{\boldsymbol{F}}\right) = \left((1+\sigma)\boldsymbol{I}_{NM} - \sigma\boldsymbol{A} \otimes \boldsymbol{B}\right)^{-1} \left(\sum_{v=1}^V \boldsymbol{w}_v \text{vec}\left(\hat{\boldsymbol{F}}^v\right)\right). \tag{26}$$

Notice that the inverse matrix on the right-hand side of Equation 26 needs too much time and memory. Therefore, we use eigen decomposed techniques to compute the approximate inverse. Let $V_A \Lambda_A V_A^T$ and $V_B \Lambda_B V_B^T$ be the eigen decomposition of the matrices $A$ and $B$, respectively. Define the matrix $U$ to be $U_{i,j} = [\Lambda_A]_{i,i} \times [\Lambda_B]_{j,j}$. By the theorem (Raymond & Kashima, 2010), the kronecker product matrix $A \otimes B$ can be eigendecomposed as $(V_A \otimes V_B) \operatorname{diag}(\operatorname{vec}(U))(V_A \otimes V_B)^T$. Then substituting it in Equation 26, we can write the inverse matrix in Equation 26 as

$$((1+\sigma) I_{NM} - \sigma A \otimes B)^{-1} = \left((1+\sigma) I_{NM} - \sigma(V_A \otimes V_B) \operatorname{diag}(\operatorname{vec}(U))(V_A \otimes V_B)^T\right)^{-1}. \tag{27}$$

Since, it holds that $(V_A \otimes V_B)(V_A \otimes V_B)^T = I_{NM}$. Equation 27 can be transformed into

$$((1+\sigma) I_{NM} - \sigma A \otimes B)^{-1} = (V_A \otimes V_B)((1+\sigma) I_{NM} - \sigma \operatorname{diag}(\operatorname{vec}(U)))^{-1}(V_A \otimes V_B)^T. \tag{28}$$

Notice that the inverse matrix in Equation 28 is a diagonal matrix whose value can be calculated as the matrix $W$

$$W_{i,j} = (1 + \sigma - \sigma U_{i,j})^{-1} \tag{29}$$

So, we can further rewrite the Equation 26 as

$$\operatorname{vec}\left(\hat{F}\right) = (V_A \otimes V_B) \operatorname{diag}(\operatorname{vec}(W))(V_A \otimes V_B)^T \left(\sum_{v=1}^{V} w_v \operatorname{vec}\left(\hat{F}^v\right)\right) \tag{30}$$

Taking out the vec-tricks operation, we can obtain the solution

$$\hat{F} = V_B \left(W \odot \left(V_B^T \left(\sum_{v=1}^{V} w_v \hat{F}^v\right) V_A\right)\right) V_A^T \tag{31}$$

$w$-subproblem: we fix all the variants except $w$. The formula is as follows:

$$\arg\min_{w} \quad \frac{1}{2} \left\|\hat{F} - \sum_{v=1}^{V} w_v \hat{F}^v\right\|_F^2 + \mu \sum_{v=1}^{V} \left(\frac{w_v}{2} \left\|F - \hat{F}^v\right\|_F^2\right) + \frac{1}{2}\beta \|w\|_2^2$$

$$s.t. \sum_{v=1}^{V} w_v = 1, w_v \geq 0, v = 1, \ldots, V. \tag{32}$$

Problem 32 can be simplified as a standard quadratic programming problem (Nocedal & Wright, 2006)

$$\arg\min_{w} \quad w^T G w - w^T h$$

$$s.t. \sum_{v=1}^{V} w_v = 1, w_v \geq 0, v = 1, \ldots, V. \tag{33}$$

where $G \in \mathbb{R}^{V \times V}$ with the element as

$$G_{i,j} = \begin{cases} \frac{1}{2}\operatorname{trace}\left(\left(\hat{F}^i\right)^T \hat{F}^j\right), & \text{if } i \neq j, \\ \frac{1}{2}\operatorname{trace}\left(\left(\hat{F}^i\right)^T \hat{F}^j\right) + \frac{1}{2}\beta, & \text{if } i = j. \end{cases} \tag{34}$$

$h$ is a vector with

$$h_i = \operatorname{trace}\left(\hat{F}^T \hat{F}^i\right) - \frac{\mu}{2} \left\|F - \hat{F}^i\right\|_F^2. \tag{35}$$

The optimization method for Equation 33 is the interior-point optimization algorithm (Byrd et al., 1999).

$\boldsymbol{\theta}_D$-**subproblem**: With the fixed all the variants except $\boldsymbol{\theta}_D$, the formula can be written as

$$\arg\min_{\boldsymbol{\theta}_D} \frac{1}{2}\sigma\text{vec}\big(\hat{\boldsymbol{F}}\big)^T \boldsymbol{L}\text{vec}\big(\hat{\boldsymbol{F}}\big)$$
$$s.t. \ \boldsymbol{L} = \boldsymbol{I}_{NM} - \big(\boldsymbol{H}_S^{-0.5}\boldsymbol{K}_S^*\boldsymbol{H}_S^{-0.5}\big) \otimes \big(\boldsymbol{H}_D^{-0.5}\boldsymbol{K}_D^*\boldsymbol{H}_D^{-0.5}\big), \tag{36}$$
$$\boldsymbol{K}_D^* = \sum_{i=1}^{P} [\boldsymbol{\theta}_D]_i^\varepsilon \boldsymbol{K}_D^i, \sum_{i=1}^{P} [\boldsymbol{\theta}_D]_i = 1, [\boldsymbol{\theta}_D]_i \geq 0, i = 1,\ldots,P.$$

Let $\boldsymbol{A} = \boldsymbol{H}_S^{-0.5}\boldsymbol{K}_S^*\boldsymbol{H}_S^{-0.5}$ and $\boldsymbol{B}^i = \boldsymbol{H}_D^{-0.5}\boldsymbol{K}_D^i\boldsymbol{H}_D^{-0.5}$. Then substituting $\boldsymbol{L}$ in Equation 36 with $\boldsymbol{A}$ and $\boldsymbol{B}^i$, the objective function 36 can be written as

$$\arg\min_{\boldsymbol{\theta}_D} -\frac{1}{2}\sigma\text{vec}\big(\hat{\boldsymbol{F}}\big)^T \sum_{i=1}^{P} \big(\boldsymbol{A} \otimes \boldsymbol{B}^i\big)\text{vec}\big(\hat{\boldsymbol{F}}\big)$$
$$s.t. \sum_{i=1}^{P} [\boldsymbol{\theta}_D]_i = 1, [\boldsymbol{\theta}_D]_i \geq 0, i = 1,\ldots,P. \tag{37}$$

Further, introduce the Lagrange multiplier $\xi$ and the objective function 37 can be converted to a Lagrange function:

$$\text{Lag}\,(\boldsymbol{\theta}_D, \xi) = -\frac{1}{2}\sigma\text{vec}\big(\hat{\boldsymbol{F}}\big)^T \sum_{i=1}^{P} \big(\boldsymbol{A} \otimes \boldsymbol{B}^i\big)\text{vec}\big(\hat{\boldsymbol{F}}\big) - \xi\left(\sum_{i=1}^{P} [\boldsymbol{\theta}_D]_i - 1\right) \tag{38}$$

Based on setting the derivative of Equation 38 *w.r.t* $\boldsymbol{\theta}_D$ and $\xi$ to zero respectively, we have the following solution

$$[\boldsymbol{\theta}_D]_i = \left(\text{vec}\big(\hat{\boldsymbol{F}}\big)^T \big(\boldsymbol{A} \otimes \boldsymbol{B}^i\big)\text{vec}\big(\hat{\boldsymbol{F}}\big)\right)^{\frac{1}{1-\varepsilon}} \bigg/ \sum_{j=1}^{P} \left(\text{vec}\big(\hat{\boldsymbol{F}}\big)^T \big(\boldsymbol{A} \otimes \boldsymbol{B}^j\big)\text{vec}\big(\hat{\boldsymbol{F}}\big)\right)^{\frac{1}{1-\varepsilon}}. \tag{39}$$

By using the vec-tricks operation, we can describe the solution as

$$[\boldsymbol{\theta}_D]_i = \text{trace}\big(\hat{\boldsymbol{F}}^T \boldsymbol{B}^i \hat{\boldsymbol{F}} \boldsymbol{A}^T\big)^{\frac{1}{1-\varepsilon}} \bigg/ \sum_{j=1}^{P} \text{trace}\big(\hat{\boldsymbol{F}}^T \boldsymbol{B}^j \hat{\boldsymbol{F}} \boldsymbol{A}^T\big)^{\frac{1}{1-\varepsilon}} \tag{40}$$

$\boldsymbol{\theta}_S$-**subproblem**:The solution of $\boldsymbol{\theta}_S$ is similarity to $\boldsymbol{\theta}_D$. Here, the optimization process is omitted and we directly give the solution

$$[\boldsymbol{\theta}_S]_i = \text{trace}\big(\hat{\boldsymbol{F}}^T \boldsymbol{B} \hat{\boldsymbol{F}} (\boldsymbol{A}^i)^T\big)^{\frac{1}{1-\varepsilon}} \bigg/ \sum_{j=1}^{Q} \text{trace}\big(\hat{\boldsymbol{F}}^T \boldsymbol{B} \hat{\boldsymbol{F}} (\boldsymbol{A}^j)^T\big)^{\frac{1}{1-\varepsilon}} \tag{41}$$

where $\boldsymbol{B} = \boldsymbol{H}_D^{-0.5}\boldsymbol{K}_D^*\boldsymbol{H}_D^{-0.5}$ and $\boldsymbol{A}^i = \boldsymbol{H}_S^{-0.5}\boldsymbol{K}_S^i\boldsymbol{H}_S^{-0.5}$.

$\boldsymbol{a}^v$-**subproblem**: By dropping all other irrelevant terms with respect $\boldsymbol{a}^v$, we have

$$\arg\min_{\boldsymbol{a}^v} \frac{1}{2}\left\|\text{vec}\big(\hat{\boldsymbol{F}}\big) - \sum_{i=1}^{V} w_i \boldsymbol{K}^i \boldsymbol{a}^i\right\|_2^2 + \mu\left(\frac{w_v}{2}\|\text{vec}\,(\boldsymbol{F}) - \boldsymbol{K}^v \boldsymbol{a}^v\|_2^2 + \frac{\lambda^v}{2}\boldsymbol{a}^{v^T}\boldsymbol{K}^v\boldsymbol{a}^v\right). \tag{42}$$

It can be observed from the objective function 42 that when training the parameter $\boldsymbol{a}^v$, other views $\boldsymbol{K}_i$ with weight $w_i$ were taken into consideration. Therefore, each partition's training is not completely separate, but involves information sharing.

Based on setting the derivative of problem 42 *w.r.t* $\boldsymbol{a}^v$ to zero, we get

$$\left(\boldsymbol{K}^v + \frac{\lambda_v}{1 + \mu w_v}\boldsymbol{I}_{NM}\right)\boldsymbol{a}^v = \frac{1}{1 + \mu w_v}\left(\text{vec}\big(\hat{\boldsymbol{F}}\big) - \sum_{i=1,i\neq v}^{V} w_i \boldsymbol{K}^i \boldsymbol{a}^i + \mu w_v \text{vec}\,(\boldsymbol{F})\right) \tag{43}$$

Let $\boldsymbol{W} = \hat{\boldsymbol{F}} - \sum_{i=1,i\neq v}^{V} \boldsymbol{w}_i \hat{\boldsymbol{F}}^i + \mu \boldsymbol{w}_v \boldsymbol{F}$, the Equation 43 can be written as

$$\boldsymbol{a}^v = \frac{1}{1+\mu\boldsymbol{w}_v} \left( \boldsymbol{K}^v + \frac{\lambda_v}{1+\mu\boldsymbol{w}_v} \boldsymbol{I}_{NM} \right)^{-1} \text{vec}\left( \boldsymbol{W} \right). \tag{44}$$

We can observe that the form of Equation 44 is similar to Equation 7. Therefore, we use eigen decomposed techniques and the vec-trick operation to effectively compute $\boldsymbol{a}^v$.

We summarize the complete optimization process for problem 24 in Algorithm 1.

---

**Algorithm 1:** Optimization for MKronRLSF-LP.

**Input:** The link matrix $\boldsymbol{F}$; The regulation parameters $\mu$, $\beta$, $\sigma$, $\varepsilon$ and $\lambda^v, v = 1, \ldots, V$;
**Output:** The predicted link matrix $\hat{\boldsymbol{F}}$;

1 Compute two sets of base kernel sets $\mathbb{K}_D$ and $\mathbb{K}_S$ by Equation 20a and 20b;
2 Initialize $\boldsymbol{a}^v, v = 1, \ldots, V$ by single view Kron-RLS; $\boldsymbol{w}_v = 1/V, v = 1, \ldots, V$; $\boldsymbol{\theta}_D^i = 1/P, i = 1, \ldots, P$;
   $\boldsymbol{\theta}_S^i = 1/Q, i = 1, \ldots, Q$;
3 **while** *Not convergence* **do**
4     Update $\hat{\boldsymbol{F}}$ by solve the subproblem 25;
5     Update $\boldsymbol{w}$ by solve the subproblem 32;
6     Update $\boldsymbol{\theta}_D$ by solve the subproblem 36;
7     Update $\boldsymbol{\theta}_S$ by Equation 41;
8     **for** $i = 1$ *to* $V$ **do**
9        Update $\boldsymbol{a}^i$ by solve the subproblem 42;
10     **end**
11 **end**

---

### A.2 Measurements

Considering that drug-side effect prediction is an extremely imbalanced classification problem and we do not want incorrect predictions to be recommended by the prediction model, we utilize the following evaluation parameters:

$$Recall = \frac{TP}{TP+FN}, \tag{45a}$$

$$Precision = \frac{TP}{TP+FP}, \tag{45b}$$

$$F_{score} = 2 \times \frac{Precision \times Recall}{Precision + Recall}, \tag{45c}$$

where $TP$, $FN$, $FP$ and $TN$ are the number of true-positive samples, false-negative samples, false-positive samples and true-negative samples, respectively. The area under the ROC curve (AUC) and area under the precision recall curve (AUPR) is also used to measure predictive accuracy, because they are the most commonly used evaluate metrics in the biomedical link prediction. The precision-recall curve shows the tradeoff between precision and recall at different thresholds. $F_{score}$ is calculated from Precision and Recall. The highest possible value of an $F_{score}$ is 1, indicating perfect precision and recall, and the lowest possible value is 0, if either precision or recall are zero. AUC can be considered as the probability that the classifier will rank a randomly chosen positive instance higher than a randomly chosen negative instance (Li et al., 2021). Therefore, we consider AUPR and $F_{score}$ more desirable metrics (Ezzat et al., 2017; Li et al., 2021).

### A.3 Baseline methods

- Best single view (BSV): Applying Kron-RLS to the best single view. The one with the maximum AUPR is chosen here.

- Committee Kron-RLS (Comm Kron-RLS)(Perrone & Cooper, 1995): Each view is trained by Kron-RLS separately, and the final classifier is a weighted average.

- Kron-RLS with Centered Kernel Alignment-based Multiple Kernel Learning (Kron-RLS+CKA-MKL)(Ding et al., 2019): Multiple kernels from the drug space and side effect space are linearly weighted by the optimized CKA-MKL. Finally, Kron-RLS is employed on optimal kernels.

- Kron-RLS with pairwise Multiple Kernel Learning (Kron-RLS+pairwiseMKL)(Cichonska et al., 2018): First, it constructs multiple pairwise kernels. Then, the mixture weights of the pairwise kernels are determined by CKA-MKL. Finally, it learns the Kron-RLS function based on the optimal pairwise kernel.

- Kron-RLS with self-weighted multiple kernel learning (Kron-RLS+self-MKL)(Nascimento et al., 2016): The optimal drug and side effect kernels are linearly weighted based on the multiple base kernel. The proper weights assignment to each kernel is performed automatically.

- Multi-view graph regularized link propagation model (MvGRLP)(Ding et al., 2021): This is an extension of the graph model (Zha et al., 2009). To fuse multi view information, multi-view Laplacian regularization is introduced to constrain the predicted values.

- Multi-view graph convolution network (MvGCN)(Fu et al., 2022): This extends the GCN (Zhang et al., 2019) from a single view to multi-view by combining the embeddings of multiple neighborhood information aggregation layers in each view.

## A.4 Code and Data Available

The code and data are available at `https://github.com/QYuQing/MKronRLSF-LP`.

## A.5 Figures

## A.6 Tables

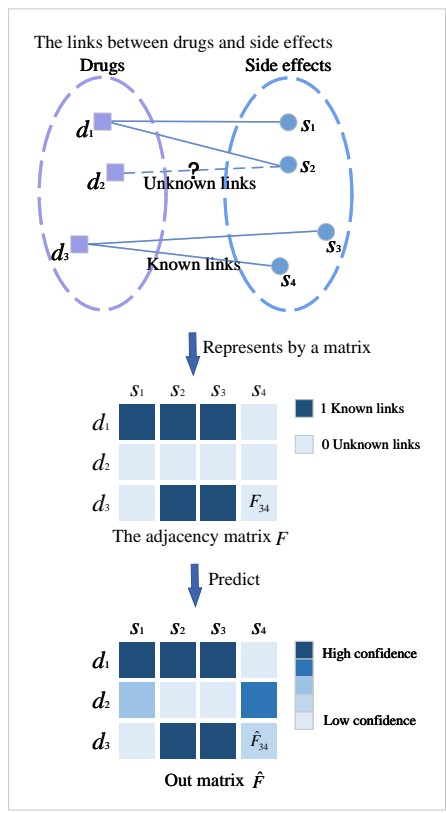

Figure 5: Visualization of drug-side effect association problems.

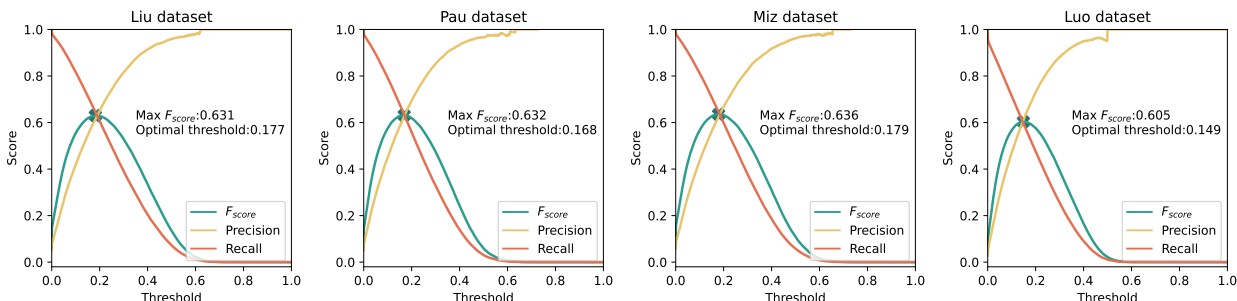

Figure 6: Results (MKronRLSF-LP) for $F_{score}$, $Recall$ and $Precision$ of different thresholds.

Table 3: Prediction performance comparison of baseline methods on four datasets.

| Dataset | Methods | AUPR(%) | AUC(%) | $Recall(\%)$ | $Precision(\%)$ | $F_{score}(\%)$ |
|---|---|---|---|---|---|---|
| Liu | BSV | 60.12±1.12 | 93.22±1.63 | 58.77±0.33 | 59.09±0.49 | 58.52±0.23 |
| | Comm Kron-RLS | 65.63±1.95 | 94.11±1.45 | 61.63±0.33 | 61.9±1.37 | 61.57±1.65 |
| | Kron-RLS +CKA-MKL | 65.92±0.43 | 92.51±0.08 | 62.11±0.43 | 63.09±0.56 | 62.59±0.41 |
| | Kron-RLS +pairwiseMKL | 62.03±0.44 | **95.01±0.06** | **65.39±0.24** | 54.46±0.30 | 59.43±0.21 |
| | Kron-RLS +self-MKL | 65.02±0.47 | 92.1±0.10 | 60.97±0.57 | **63.12±0.61** | 62.03±0.52 |
| | MvGRLP | 66.32±0.45 | 94.29±0.08 | 63.56±0.46 | 60.87±0.62 | 62.18±0.39 |
| | MvGCN | 62.69±1.81 | 94.01±0.87 | 60.81±0.37 | 60.33±1.31 | 60.48±1.15 |
| | MKronRLSF-LP | **68.02±0.44** | 94.78±0.13 | 65.18±0.93 | 61.27±1.08 | **63.02±0.43** |
| Pau | BSV | 65.26±0.98 | 94.57±0.34 | 62.54±0.5 | 60.77±1.27 | 60.65±0.73 |
| | Comm Kron-RLS | 65.63±0.36 | 94.78±0.13 | 64.01±0.38 | 60.05±0.49 | 61.01±0.27 |
| | Kron-RLS +CKA-MKL | 65.49±0.37 | 92.39±0.13 | 61.65±0.40 | **63.22±0.51** | 62.42±0.27 |
| | Kron-RLS +pairwiseMKL | 63.48±0.39 | **95.02±0.07** | **78.1±0.26** | 45.01±0.48 | 57.11±0.36 |
| | Kron-RLS +self-MKL | 64.11±1.75 | 91.94±0.25 | 62.37±0.29 | 60.97±1.57 | 61.65±0.79 |
| | MvGRLP | 66.17±0.32 | 94.42±0.07 | 62.18±0.38 | 61.95±0.45 | 62.06±0.22 |
| | MvGCN | 63.51±1.43 | 94.08±0.49 | 63.21±0.69 | 57.94±1.34 | 60.4±1.78 |
| | MKronRLSF-LP | **67.81±0.37** | 94.81±0.18 | 65.72±3.58 | 60.65±3.75 | **62.87±0.48** |
| Miz | BSV | 56.58±2.33 | 90.71±2.06 | 62.76±0.69 | 53.94±2.31 | 55.39±2.33 |
| | Comm Kron-RLS | 58.08±1.07 | 91.36±1.25 | 62.37±0.81 | 55.16±1.99 | 56.54±1.77 |
| | Kron-RLS +CKA-MKL | 66.92±0.44 | 92.58±0.14 | 62.62±0.52 | **64.3±0.46** | 61.45±0.44 |
| | Kron-RLS +pairwiseMKL | 62.13±0.29 | **94.70±0.11** | 63.78±0.47 | 56.26±0.42 | 59.79±0.30 |
| | Kron-RLS +self-MKL | 65.84±0.43 | 92.06±0.16 | 63.63±0.48 | 61.77±0.52 | 60.68±0.43 |
| | MvGRLP | 66.68±0.35 | 94.10±0.12 | 63.46±0.43 | 61.82±0.30 | 62.63±0.29 |
| | MvGCN | 62.17±1.90 | 93.35±1.73 | 59.54±0.43 | 60.74±1.78 | 59.76±1.95 |
| | MKronRLSF-LP | **68.35±0.38** | 94.47±0.09 | **65.15±2.77** | 62.10±3.19 | **63.45±0.53** |
| Luo | BSV | 60.40±0.40 | 94.40±0.11 | 58.28±0.41 | 58.68±0.46 | 58.48±0.39 |
| | Comm Kron-RLS | 54.19±1.36 | 91.92±4.01 | 57.64±2.46 | 53.16±1.97 | 52.99±1.54 |
| | Kron-RLS +CKA-MKL | 60.87±0.36 | 92.03±0.15 | 55.55±0.34 | **64.15±0.46** | 59.54±0.36 |
| | Kron-RLS +pairwiseMKL | 50.29±0.29 | 94.37±0.10 | 55.66±0.39 | 45.97±0.39 | 50.35±0.31 |
| | Kron-RLS +self-MKL | 22.29±1.57 | 79.74±1.62 | 56.62±1.47 | 20.91±1.64 | 28.23±1.15 |
| | MvGRLP | 61.76±0.45 | 94.08±0.07 | 58.70±0.40 | 60.05±0.61 | 58.37±0.42 |
| | MvGCN | 61.18±0.41 | **94.54±0.1** | 57.94±0.37 | 61.26±0.48 | 51.07±0.38 |
| | MKronRLSF-LP | **63.32±0.58** | 94.07±0.14 | **59.43±0.95** | 61.58±1.22 | **60.47±0.39** |

Table 4: The optimal parameters $\lambda^v$ obtained with the single view Kron-RLS model (based on the relative pairwise kernel).

| $\otimes$ | $\boldsymbol{K}_{GIP,S}$ | $\boldsymbol{K}_{GIP,S}$ | $\boldsymbol{K}_{GIP,S}$ | $\boldsymbol{K}_{GIP,S}$ | $\boldsymbol{K}_{GIP,S}$ |
|---|---|---|---|---|---|
| $\boldsymbol{K}_{GIP,D}$ | $2^0$ | $2^2$ | $2^2$ | $2^1$ | $2^1$ |
| $\boldsymbol{K}_{COS,D}$ | $2^2$ | $2^3$ | $2^3$ | $2^2$ | $2^{-2}$ |
| $\boldsymbol{K}_{Corr,D}$ | $2^3$ | $2^3$ | $2^4$ | $2^2$ | $2^4$ |
| $\boldsymbol{K}_{MI,D}$ | $2^0$ | $2^1$ | $2^1$ | $2^0$ | $2^{-1}$ |
| $\boldsymbol{K}_{NTK,D}$ | $2^2$ | $2^4$ | $2^3$ | $2^1$ | $2^1$ |

Table 5: Summary of the threshold of baseline methods on four datasets.

| Methods | Liu | Pau | Miz | Luo |
|---|---|---|---|---|
| BSV | 0.145 | 0.146 | 0.142 | 0.128 |
| Comm Kron-RLS | 0.205 | 0.204 | 0.192 | 0.183 |
| Kron-RLS+CKA-MKL | 0.100 | 0.106 | 0.099 | 0.102 |
| Kron-RLS+pairwiseMKL | 0.149 | 0.159 | 0.101 | 0.107 |
| Kron-RLS+self-MKL | 0.119 | 0.116 | 0.113 | 0.129 |
| MvGRLP | 0.090 | 0.091 | 0.094 | 0.085 |
| MVGCN | 0.225 | 0.237 | 0.208 | 0.197 |
| MKronRLSF-LP | 0.177 | 0.168 | 0.179 | 0.149 |

Table 6: Prediction performance comparison of other drug-side effect predictors on Liu datasets.

| Methods | AUPR(%) | AUC(%) | $Recall$(%) | $Precision$(%) | $F_{score}$(%) |
|---|---|---|---|---|---|
| Liu's method | 28.0 | 90.7 | **67.5** | 34.0 | 45.2 |
| Cheng's method | 59.2 | 92.2 | 59.0 | 55.7 | 56.9 |
| RBMBM | 61.6 | 94.1 | 61.5 | 57.4 | 59.4 |
| INBM | 64.1 | 93.4 | 60.7 | 60.4 | 60.6 |
| Ensemble model | 66.1 | 94.8 | 62.3 | 61.1 | 61.7 |
| MKL-LGC[a] | 67.0 | **95.1** | - | - | - |
| NDDSA with sschem[c] | 60.5 | 94.1 | 57.9 | 56.4 | 57.1 |
| NDDSA without sschem[c] | 60.4 | 94.0 | 57.4 | 56.8 | 57.1 |
| MKronRLSF-LP | **68.2** | 94.7 | 63.8 | **62.5** | **63.1** |

- represents not available; the bold and underlined values represent the best and second performance measure in each column, respectively;
a and b represents the results are derived from (Ding et al., 2018) and (Shabani-Mashcool et al., 2020), respectively.

Table 7: Prediction performance comparison of other drug-side effect predictors on Pau datasets.

| Methods | AUPR(%) | AUC(%) | $Recall(\%)$ | $Precision(\%)$ | $F_{score}(\%)$ |
|---|---|---|---|---|---|
| Pau's method[a] | 38.9 | 89.7 | 51.7 | 36.1 | 42.5 |
| Liu's method | 34.7 | 92.1 | **64.6** | 40.0 | 49.5 |
| Cheng's method | 58.8 | 82.3 | 58.3 | 55.0 | 56.6 |
| RBMBM | 61.3 | 94.1 | 60.8 | 57.7 | 59.2 |
| INBM | 64.1 | 93.4 | 60.8 | 60.5 | 60.7 |
| Ensembel model | 66.0 | 94.9 | 62.4 | 61.2 | 61.6 |
| MKL-LGC[b] | 66.8 | **95.2** | - | - | - |
| NDDSA with sschem[c] | 60.3 | 94.2 | 59.3 | 54.9 | 57.0 |
| NDDSA without sschem[c] | 60.3 | 94.1 | 58.2 | 55.9 | 57.0 |
| MKronRLSF-LP | **67.9** | 94.7 | 63.4 | **62.9** | **63.2** |

- represents not available; the bold and underlined values represent the best and second performance measure in each column, respectively;

a, b and c represents the results are derived from (Zhang et al., 2016), (Ding et al., 2018) and (Shabani-Mashcool et al., 2020), respectively.

Table 8: Prediction performance comparison of other drug-side effect predictors on Miz datasets.

| Methods | AUPR(%) | AUC(%) | $Recall(\%)$ | $Precision(\%)$ | $F_{score}(\%)$ |
|---|---|---|---|---|---|
| Miz's method[a] | 41.2 | 89.0 | 52.7 | 38.7 | 44.6 |
| Liu' method | 36.3 | 91.8 | **64.0** | 41.5 | 50.5 |
| Cheng's method | 56.0 | 92.3 | 58.4 | 56.8 | 57.6 |
| RBMBM | 61.7 | 93.9 | 60.5 | 58.8 | 59.6 |
| INBM | 64.6 | 93.2 | 61.6 | 60.5 | 61.1 |
| Ensemble model | 66.6 | 94.6 | 62.4 | 61.9 | 62.2 |
| MKL-LGC[b] | 67.3 | **94.8** | - | - | - |
| NDDSA with sschem[c] | 60.6 | 93.9 | 58.8 | 56.3 | 57.5 |
| NDDSA without sschem[c] | 60.7 | 93.6 | 60.0 | 55.5 | 57.6 |
| MKronRLSF-LP | **68.5** | 94.5 | 63.0 | **64.2** | **63.6** |

- represents not available; the bold and underlined values represent the best and second performance measure in each column, respectively;

a, b and c represents the results are derived from (Zhang et al., 2016), (Ding et al., 2018) and (Shabani-Mashcool et al., 2020), respectively.

Table 9: Prediction performance comparison of other drug-side effect predictors on Luo datasets.

| Methods | AUPR(%) | AUC(%) | $Recall(\%)$ | $Precision(\%)$ | $F_{score}(\%)$ |
|---|---|---|---|---|---|
| Liu's method | 39.4 | 93.5 | **59.6** | 48.3 | 53.3 |
| Cheng's method | 53.2 | 90.9 | 53.1 | 52.3 | 52.7 |
| RBMBM | 55.1 | 93.5 | 56.1 | 54.3 | 55.1 |
| INBM | 57.3 | 91.7 | 55.8 | 56.7 | 56.2 |
| Ensemble model | 58.6 | 93.9 | 46.1 | **68.4** | 55.1 |
| MKL-LGC | 61.7 | 94.6 | - | - | - |
| NDDSA with sschem[a] | 53.1 | 94.2 | 47.6 | 57.3 | 52.0 |
| NDDSA without sschem[a] | 44.5 | 93.7 | 44.7 | 47.8 | 46.2 |
| GCRS[b] | 27.2 | **95.7** | - | - | - |
| SDPred | 22.6 | 94.6 | - | - | - |
| MKronRLSF-LP | **63.5** | 94.1 | 59.2 | 61.9 | **60.5** |

- represents not available; the bold and underlined values represent the best and second performance measure in each column, respectively;

a,b represents the results are derived from (Shabani-Mashcool et al., 2020) and (Xuan et al., 2022), respectively.

