# OpenReview forum: "Multiple Kronecker RLS fusion-based link propagation for drug-side effect prediction"
_TMLR — Accepted by TMLR_

### Review · Reviewer_8TLU · 2024-04-15

**Summary Of Contributions:**

This paper proposes a new multi-view learning method called MKronRLSF-LP for predicting drug-side effects, which is an important problem in pharmacovigilance. The method extends the Kronecker regularized least squares (Kron-RLS) framework by learning a consensus partition from multiple views. It also incorporates multiple graph Laplacian regularization to enhance the performance. An efficient optimization algorithm is developed to fuse the multiple Kron-RLS submodels and make predictions without explicitly computing large pairwise matrices. Experiments on datasets demonstrate the good performance of MKronRLSF-LP compared to various baseline methods and state-of-the-art approaches.

**Audience:**

Yes

**Broader Impact Concerns:**

I didn't see any potential concerns.

**Claims And Evidence:**

Yes

**Requested Changes:**

1. This paper should carefully discuss the compare the results with the previous graph neural network based method.

2. The authors should improve the overall presentation of the paper. This includes ensuring proper citation formatting throughout the manuscript, improving the grammar and sentence structures.

3. The discussion of experiments are also weak. The authors should provide more understandings and ablation studies of proposed methods.

**Strengths And Weaknesses:**

Strengths:
1. This paper focuses on an important problem.
2. The method is computationally efficient.
3. This paper achieves good performance on the given benchmarks.

Weakness:
1. This work seems combine the previous method such as Kronecker regularized least squares (Kron-RLS) and Multi-view graph regularized link propagation model (MvGRLP), etc. Based on the results, they are incremental improvements on the previous method.
2. The presentation and citation should be improved.

---

> ### Author Response · Authors · 2024-05-12
>
> We thank the reviewer for the careful reading and the valuable comments. We extend the Kron-RLS model to the multi-view setting by the consensus partition and multiple graph Laplacian regularization. The main difference between our method and previous methods （such as Kron-RLS+CKA-MKL，Kron-RLS+pairwiseMKL and MvGRLP） appears in the multi-view setting.
>
> >1.  This paper should carefully discuss the compare the results with the previous graph neural network based method.
>
> Thanks for your comments. We have discussed the comparison results with MvGCN.
>
> MvGCN [1] is worse ranked than our method (Figure 3). Another point worth mentioning is that there is no sufficient statistical evidence to support that MvGCN performs better than model-based methods. MvGCN uses shallow GCN to avoid over-smoothing. The shallow GCN [2] can only capture local neighbourhood information of nodes, but the global features of the network have not been fully explored. A result of this is inaccurate embedding vectors.
>
> Some of above sentences have been added in section “Comparison with baseline methods”.
>
> When MvGCN deal with Luo dataset, its running time exceeds 2 hours. This is because MvGCN utilizes a self-supervised learning strategy based on deep graph infomax (DGI) [3] to initialize node embeddings. Whenever there are many nodes in a bipartite network, DGI takes a very long time to implement.
>
> Some of above sentences have been added in section “Comparisons of computational speed”.
>
>  - [1] Haitao Fu, Feng Huang, Xuan Liu, Yang Qiu, and Wen Zhang. Mvgcn: data integration through multi-view graph convolutional network for predicting links in biomedical bipartite networks. Bioinformatics, 38(2): 426–434, 2022.
>  - [2] Xupeng Miao, Wentao Zhang, Yingxia Shao, Bin Cui, Lei Chen, Ce Zhang, and Jiawei Jiang. Lasagne: A multi-layer graph convolutional network framework via node-aware deep architecture. IEEE Transactions on Knowledge and Data Engineering, 35(2):1721–1733, 2021.
>  - [3] Veličković P, Fedus W, Hamilton W L, et al. Deep graph infomax[J]. arXiv preprint arXiv:1809.10341, 2018.
>
> >2. The authors should improve the overall presentation of the paper. This includes ensuring proper citation formatting throughout the manuscript, improving the grammar and sentence structures.
>
> Thanks for your comments. We have improved the overall presentation of the paper. In addition, the paper has undergone editing by Elsevier Language Editing Services to further enhance its quality.
>
> >3.  The discussion of experiments are also weak. The authors should provide more understandings and ablation studies of proposed methods.
>
> Thanks for your comments. We have added new experiments to section "Experiments", including ablation studies and computation speed comparisons.

---

### Review · Reviewer_rgQm · 2024-04-21

**Summary Of Contributions:**

The authors propose to tackle the problem of drug-side effect prediction which is a link prediction problem or a recommendation problem. They propose to use a multiple kernel approach. They start from a Kronecker regularised least square formulation and add two contributions: a notion of consensus partition and a regularisation via a multiple graph Laplacian constraint.

**Audience:**

Yes

**Claims And Evidence:**

No

**Requested Changes:**

Please address the previous issues:
1. explain the notion of partition and consensus (perhaps adding a visual support) in plain English before using any formal notation
2. justify intuitively the reasoning on why these changes (partition and Laplace smoothing) should in principle help here and how these ideas could be useful in related problems (explain which are the characteristics that these problems should have to benefit from your approach)
3. provide a) an ablation study, b)  critical differences diagrams, c) efficiency comparisons,  d) artificial problem

**Strengths And Weaknesses:**

The main issue with the current paper is in the 1) presentation, 2) justification and 3) validation.
1. the authors never explain the notion of partition or consensus, which is the main contribution of the approach. This prevents the reader from understanding what is the gist of the proposal and why this particular approach should yield an advantage.
2. The authors very briefly mention at page 7  that  "the node pair sharing the same labels should be kept close together. This means any two pair nodes with same labels have a higher similarity. That is, the pair node labels with stronger correlations are assumed to be closer to each other in the original space. This is analogous to the Laplacian operator on manifolds". This is the only justification for the introduction of the laplacian regularisation offered to the reader. No justification is offered for the partition notion (which is not explained).
3. a more thorough empirical validation should be offered: a) an ablation experimental design should be employed to show the relative importance of the partition notion and the laplacian regularisation; b) a significance test is missing, so we do not know if the improvement reported is statistically significative (please use a post-hoc analysis and a critical differences diagram); c) the author says that "we avoid explicit computation of any pairwise matrices, which makes our method suitable for solving problems in large pairwise spaces", but do not offer an efficiency comparison with other approaches; d) the authors should use an artificial problem where the difficulty of the problem is controlled by design and show the regime when the proposed approach works and when it stops working (e.g. when the number of missing links is above a certain threshold w.r.t. the size of the network, etc)

---

> ### Author Response · Authors · 2024-05-12
>
> We thank the reviewer for the careful reading and the valuable comments.
> >1.  explain the notion of partition and consensus (perhaps adding a visual support) in plain English before using any formal notation
>
> We have explained the notion of partition and consensus in plain English. In order for readers to have a clearer understanding of multi-view methods, we illustrate our taxonomy of multi-view learning method literature in Figure 1.
>
> "Partition" commonly refers to the learned result. This concept is more commonly found in classification and clustering tasks[1-3]. For example, the clustering matrix $H$ is also called the partition matrix in [3]. In our study, we call the learned prediction scores $\hat F ^v$ as partition, $\hat F$ as consensus partition.
>
> In multi-view methods, the consensus principle establishes consistency between partitions from different views [4-6]. For example, Liu et al. [7] finds a consensus clustering matrix $H$ from $H ^v,v=1...V$ by the objective function:
>
> $\mathop {\max }\limits_H {\rm{Tr}}\left[ {{H^T}\left( {\sum\limits_{v = 1}^V {{H_v}{W_v}} } \right)} \right]$.
>
> In our method, the consensus partition $\hat F$ is a linear combination of other partitions $\hat F ^v,v=1...V$.
>
> Some of above sentences have been added in section “Introduction” and "The MKronRLSF-LP model".
>
>  - [1]J. Liu, X. Liu, Y. Yang, Q. Liao, Y. Xia, Contrastive multi-view kernel learning, IEEE Transactions on Pattern Analysis and Machine Intelligence (2023).
>  - [2]E. Bruno, S. Marchand-Maillet, Multiview clustering: a late fusion approach using latent models, in: Proceedings of the 32nd international ACM SIGIR conference on Research and development in information retrieval, 2009, pp. 736–737.
>  - [3]S. Wang, X. Liu, E. Zhu, C. Tang, J. Liu, J. Hu, J. Xia, J. Yin, Multi-view clustering via late fusion alignment maximization., in: IJCAI, 2019, pp. 3778–3784.
>  - [4]S. Huang, Y. Liu, I. W. Tsang, Z. Xu, J. Lv, Multi-view subspace clustering by joint measuring of consistency and diversity, IEEE Transactions on Knowledge and Data Engineering (2022).
>  - [5]X. Jia, X.-Y. Jing, X. Zhu, S. Chen, B. Du, Z. Cai, Z. He, D. Yue, Semi-supervised multi-view deep discriminant representation learning, IEEE transactions on pattern analysis and machine intelligence 43 (7) (2020) 2496–2509.
>  - [6]K. Zhan, F. Nie, J. Wang, Y. Yang, Multiview consensus graph clustering, IEEE Transactions on Image Processing 28 (3) (2018) 1261–1270.
>  - [7]X. Liu, X. Zhu, M. Li, L. Wang, C. Tang, J. Yin, D. Shen, H. Wang, W. Gao, Late fusion incomplete multi-view clustering, IEEE transactions on pattern analysis and machine intelligence 41 (10) (2018) 2410–2423.

---

> > ### Author Response · Authors · 2024-05-12
> >
> > >2. justify intuitively the reasoning on why these changes (partition and Laplace smoothing) should in principle help here and how these ideas could be useful in related problems (explain which are the characteristics that these problems should have to benefit from your approach)
> >
> > We have rewritten the motivations for using these changes in section "The MKronRLSF-LP model".
> >
> > In multi-view methods, the consensus principle establishes consistency between partitions from different views. However, it's essential to find that these partitions deliver varying degrees of importance to the final prediction, unlike fusion without discrimination. To facilitate this, we introduce a consensus partition, denoted by $\hat F$. It is a weighted linear combination of partitions $\hat F_v$ from multiple distinct views. A variable $w _v$ is introduced for view $v$ which characterizes its importance, which is calculated based on the training error. To prevent sparse situations, we employ $\left\|  \cdot  \right\|_2^2$ to smooth the weights.
> >
> > In Equation 23, we observe that the consensus partition $\hat F$ fits to an adjacency matrix $F$ by an indirect path. As described in section "Problem description", false zeros represent unobserved links in the network. Hence, we must avoid overfitting the observed matrix $F$. Inspired by manifold scenarios, the Laplacian operator adeptly mitigates overfitting and noise, preserving the original data structure and keeping nodes with common labels closely associated. This approach is simple, and empirical evidence confirms its effective performance [1-3]. Here, we apply multiple graph Laplacian regularization to Equation 23, which can effectively explore multiple different views and boost the performance of $\hat F$. Specifically, the Kronecker product Laplacian matrix is calculated from the optimal drug and side effect similarity matrix, which are weighted linear combinations of multiple related kernel matrices. The weight of each kernel can be adaptively optimized during the training process and reduce the impact of noisy or less relevant graphs.
> >
> >  - [1]J. Pang, G. Cheung, Graph laplacian regularization for image denoising: Analysis in the continuous domain, IEEE Transactions on Image Processing 26 (4) (2017) 1770–1785.
> >  - [2]G. Chao, S. Sun, Semi-supervised multi-view maximum entropy discrimination with expectation laplacian regularization, Information Fusion 45 (2019) 296–306.
> >  - [3]B. Jiang, C. Zhang, Y. Zhong, Y. Liu, Y. Zhang, X. Wu, W. Sheng, Adaptive collaborative fusion for multi-view semi-supervised classification, Information Fusion 96 (2023) 37–50.

---

> > > ### Author Response · Authors · 2024-05-12
> > >
> > > >3.  provide a) an ablation study, b) critical differences diagrams, c) efficiency comparisons, d) artificial problem We have made the following changes
> > >
> > >  -  a). To validate the benefits of jointly applying the consensus partition and multiple graph Laplacian constraint, we conduct an ablation study by excluding a particular component. The results are represented in section "Ablation study".
> > >
> > >  -  b). We use post-hoc test statistics to statistically assess the different metrics shown in Table 3. Figure 3 shows the results of these tests visualized as Critical Difference diagrams. These results show that MKronRLSF-LP is significantly better ranked than all methods in terms of AUPR, $Recall$ and $F_{score}$.
> > >
> > >  -  c). In order to demonstrate the effectiveness of MKronRLSF-LP, we compare it to different baseline methods in terms of computational speed. Except MvGCN, other methods are performed on a PC equipped with an Intel Core i7-13700 and 16GB RAM. Because MvGCN is a deep learning-based method, it is performed on a workstation equipped with a NVIDIA GeForce RTX 3090 GPU. For all baseline methods, we tested 10 times to report the mean running time. The results are shown in Table 2.
> > >
> > >  -  d). We truly appreciate your suggestions for testing our proposed approach by designing an artificial problem. Indeed, that could potentially provide a visually interesting presentation of our work’s variability with respect to the complexity of the problem. However, throughout this study, we have used real-world datasets to drive our research. The datasets used are derived from trusted, previously validated databases that have been widely used in various studies. Moreover, to the best of our knowledge there is no tool for simulating biological link networks at present. We look forward to your further suggestions on this issue.

---

### Review · Reviewer_HTZM · 2024-04-29

**Summary Of Contributions:**

The authors propose to extend the Kronecker regularized least squares (Kro-RLS) technique to classify drug-side effect. Their proposal focuses on late fusion by finding a consensus partition, applying multiple graph Laplacian regularizations. To accomplish this, they introduce an optimization algorithm and subsequently contrast their approach with various baseline methods.

**Audience:**

Yes

**Broader Impact Concerns:**

There are no ethical concerns associated with the release of this work. However, providing a model card would facilitate the method's proper utilization.

**Claims And Evidence:**

No

**Requested Changes:**

## Majors requests
- The proposed technique aims to determine the optimal threshold by maximizing the F-score, which is also one the main metrics used for comparing with alternative approaches. However, the absence of a validation or test split raises concerns about potential overfitting.

- Various methods listed in Tables 4, 5, and 6, such as "RBMBM," "MKL-LGC," and "NDDSA," are not previously introduced in the manuscript. Providing brief descriptions for these methods would enhance the reader's understanding and facilitate a more informed assessment.

- In Section 6, the final paragraph appears disconnected from the preceding content: "MKronRLSF-LP uses L2 loss to measure the quality of approximation. However, the L2 loss function measure is sensitive to outliers and the interaction matrix contains false zeros. Thus, feature work contain replaces the quadratic form of residues by robust loss, such as LINEX lossTang et al. (2021) and p-power lossChen et al. (2017)".

- In Section 5.6, the following sentence seems to be exaggerated given that AUC results are reported yet not considered: "Obviously, MKronRLSF-LP achieves the best prediction results for all datasets (the highest AUPR and Fscore score on all datasets)".


## Minor requests
- There are numerous typos and oversights present in the manuscript: "Based on MKL method, Ding et al. Ding et al. (2019) and Nascimento et al. Nascimento et al. (2016)", "Psecision".

- The description of the datasets is nearly absent, which could hinder readers' understanding of the data used in the study.

- In Table 1, "Zero of rates" appears to be nonsensical and may require clarification or correction.

**Strengths And Weaknesses:**

## Strengths
- Introduce a novel late fusion technique, including various optimization steps, and contrast that to early fusion methods.
- The methodology and rationale are presented comprehensively, facilitating understanding.
- Comparative analysis with multiple Kron-based techniques and other multi-view graph approaches, as well as deep learning based approaches, strengthens the study's contextualization.

## Weaknesses
- There's a lack of clarity on how potential overfitting is mitigated and how the approach is fairly evaluated.
- Certain claims may be somewhat exaggerated or require more substantiation.
- A thorough proofreading pass would enhance the manuscript's clarity and readability.

---

> ### Author Response · Authors · 2024-05-12
>
> We thank the reviewer for the careful reading and the valuable comments.
> >#### Majors requests 1.
> >The proposed technique aims to determine the optimal threshold by maximizing the F-score, which is also one the main metrics used for comparing with alternative approaches. However, the absence of a validation or test split raises concerns about potential overfitting.
>
> We realize that we may not have detailed our threshold finding process thoroughly.
>
> We'd like to clarify that our process does indeed include a form of validation. For a certain validation set in the five fold cross-validation procedure, we collect the labels and their corresponding predicted scores. Then, we obtain the optimal threshold by maximizing the $F_{score}$ on the predicted scores and labels from this validation sets. After obtaining this threshold, we apply it to a 5-CV process with different cross indexes. Run this 10 times and report the mean and standard deviation of $F_{score}$.
>
> Some of above sentences have been added in section “Threshold finding”.
>
> Other methods use similar strategies to find optimal thresholds [1-3]. For example, for each validation set in the tenfold cross-validation procedure, Diego et al. [1] collected the labels and their corresponding predicted scores. Then, thresholds are set by maximum likelihood using likelihood density functions for each class that are estimated during cross-validation.
>
>  - [1] D. Galeano, S. Li, M. Gerstein, A. Paccanaro, Predicting the frequencies of drug side effects, Nature communications 11 (1) (2020) 1–14.
>  - [2] W. Zhang, H. Zou, L. Luo, Q. Liu, W. Wu, W. Xiao, Predicting potential side effects of drugs by recommender methods and ensemble learning, Neurocomputing 173 (2016) 979–987.
>  - [3] Y. Ding, J. Tang, F. Guo, Identification of drug-side effect association via multiple information integration with centered kernel alignment, Neurocomputing 325 (2019) 211–224. 10.028.
>
> >#### Majors requests 2.
> >Various methods listed in Tables 4, 5, and 6, such as "RBMBM," "MKL-LGC," and "NDDSA," are not previously introduced in the manuscript. Providing brief descriptions for these methods would enhance the reader's understanding and facilitate a more informed assessment.
>
> We have summarized these methods (including Pau's method, Miz's method, Liu's method, Cheng's method, RBMBM, NNDSA and MKL-LGC) in section "Introduction ".
>
> In their work, Pauwels et al. predicted the side effects of drugs (Pau's method) by applying K-nearest neighbor (KNN), support vector machine (SVM), ordinary canonical correlation analysis (OCCA) and sparse canonical correlation analysis (SCCA) from drug chemical substructures; furthermore, their experiment outcome suggests that SCCA performs the best. Mizutani et al. utilized SCCA to associate targeted proteins with side effects (Miz's method). Liu et al. predicted drug side effects (Liu's method) using SVM and multivariate information, such as the phenotypic characteristics, chemical structures, and biological properties of the drug. Cheng et al. proposed a phenotypic network inference classifier to associate drugs with side effects (Cheng's method). In Zhang et al., an ensemble model based on Liu et al.'s method, Cheng et al.'s method, the integrated neighbour-based method (INBM) and restricted Boltzmann machine-based method (RBMBM) was provided. The authors found that their ensemble method achieved the highest classification accuracy. NDDSA models the drug-side effects prediction problem using a bipartite graph and applies a resource allocation method to find new links. MKL-LGC integrates multiple kernels to describe the diversified information of drugs and side-effects. These kernels are then combined using an optimized linear weighting algorithm. The Local and Global Consistency algorithm (LGC) is used to estimate new potential associations based on the integrated kernel information.
>
> Some of above sentences have been added in section “Introduction”.
>
> >Majors requests 3.
> >In Section 6, the final paragraph appears disconnected from the preceding content: "MKronRLSF-LP uses L2 loss to measure the quality of approximation. However, the L2 loss function measure is sensitive to outliers and the interaction matrix contains false zeros. Thus, feature work contain replaces the quadratic form of residues by robust loss, such as LINEX lossTang et al. (2021) and p-power lossChen et al. (2017)".
>
> Thanks for your comments. We have removed that portion entirely. And, we've changed the title of the section from "Conclusion and future work" to simply "Conclusion".

---

> > ### Author Response · Authors · 2024-05-12
> >
> > >#### Majors requests 4.
> > >In Section 5.6, the following sentence seems to be exaggerated given that AUC results are reported yet not considered: "Obviously, MKronRLSF-LP achieves the best prediction results for all datasets (the highest AUPR and Fscore score on all datasets)".
> >
> > We have changed "Obviously, MKronRLSF-LP achieves the best prediction results for all datasets (the highest AUPR and Fscore score on all datasets)" to "Obviously, MKronRLSF-LP achieves the highest AUPR and $F_{score}$ on all datasets. In the problem of drug-side effects prediction, AUPR and $F_{score}$ more desirable metrics. Therefore, we conclude that our method outperforms the other assessed methods."
> >
> > Identification of drug-side effects is an example of the link prediction problem. The AUPR and AUC metrics are commonly used outcome metrics in biomedical link predictions. In link prediction problem, the links between nodes is highly sparse. In practice, we do not want incorrect predictions to be recommended by prediction models[1]. In this case, Precision and Recall are valuable measures for successful predictions. The precision-recall curve shows the trade off between precision and recall at different thresholds. $F_{score}$ is calculated from Precision and Recall. The highest possible value of an $F_{score}$ is 1, indicating perfect precision and recall, and the lowest possible value is 0, if either precision or recall are zero. AUC can be considered as the probability that the classifier will rank a randomly chosen positive instance higher than a randomly chosen negative instance [2]. So in our context, we consider AUPR and $F_{score}$ more desirable metrics[1,2]. Our method achieves the highest AUPR and $F_{score}$ on all datasets, which led us to conclude that it outperforms the other assessed methods.
> >
> > Some of above sentences have been added in section “Measurements”.
> >
> > >#### Minor requests 1
> > >There are numerous typos and oversights present in the manuscript: "Based on MKL method, Ding et al. Ding et al. (2019) and Nascimento et al. Nascimento et al. (2016)", "Psecision".
> >
> > We have carefully checked the entire manuscript again and corrected these, and any similar errors.
> >
> > >#### Minor requests 2
> > >The description of the datasets is nearly absent, which could hinder readers' understanding of the data used in the study.
> >
> >  We have added descriptions of relevant datasets to section “Dataset”.
> >
> > Four real drug-side effect datasets are used to assess the effectiveness of our proposed method. Pau dataset is derived from the SIDER database which contains information about drugs and their recorded side effects. Miz dataset includes information about drug-protein interactions and drug-side effect interactions, obtained from the DrugBank and SIDER database, respectively. There were 658 drugs with both targeted protein and side effect information. Additionally, Liu et al. mapped drugs in SIDER to DrugBank 3.0, resulting in a final dataset of 832 drugs and 1385 side effects. Luo dataset has a large number of side effects and was extracted from the SIDER 2.0.
> >
> > Some of above sentences have been added in section “Dataset”.
> >
> > >#### Minor requests 3
> > >-   In Table 1, "Zero of rates" appears to be nonsensical and may require clarification or correction.
> >
> > We have amended this term to “Sparsity,” which more accurately represents the
> > concept we wish to describe.
> >
> > For clarity, I would like to correct my previous statement: in the adjacency matrix we used in this study, a zero entry indeed signifies an unknown link, not merely the absence of an link. Therefore, the term initially used, "Zero of rates", was intended to reflect the sparsity of known links within this matrix. With your insightful comment, we have replaced this term with "Sparsity", which we believe more accurately conveys this concept.

---

### Decision · Action_Editor_ijJN · 2024-06-09

**Recommendation:** Accept as is

**Comment:**

All reviewers were supportive of accepting the paper. The Authors improved the work significantly during the rebuttal phase, in particular by adding several clarifications and the ablation study. The emphasized strengths of the paper included novelty of the model and the comprehensive evaluation.

**Audience:**

The findings of the paper will be clearly interesting to the subcommunity interested in link prediction methods, as well as specifically predicting the side effects of drugs. It is worth also mentioning that the hybrid approach used for fusing information from different views, which combines the strengths of early and late fusion, might serve as a high level inspiration for other multimodal tasks.

**Claims And Evidence:**

The paper proposes MKronRLSF-LP, a novel method for the link prediction task. The core claim is that the method achieves state-of-the-art results on standard benchmarks for predicting drug side effects.

The area chair is not an expert in this research field. However, reviewers more knowlegeable with the benchmarks and methods used in this field were unanimously supportive of the methodology used by the paper and found the experiments convincing. Certain issues raised during the review phase were largely resolved in the rebuttal (in particular, the added ablation brings clarity to the importance of consensus partition and laplacian regularization).